# Representation-Level Counterfactual Calibration for Debiased Zero-Shot Recognition

**Pei Peng[1]    Ming-Kun Xie[1]    Hang Hao[1]    Tong Jin[1]    Sheng-Jun Huang[1]\***

[1] Nanjing University of Aeronautics and Astronautics, Nanjing, China

{pengpei,mkxie,haohang,tongjin,huangsj}@nuaa.edu.cn

## Abstract

Object–context shortcuts remain a persistent challenge in vision-language models, undermining zero-shot reliability when test-time scenes differ from familiar training co-occurrences. We recast this issue as a causal inference problem and ask: *Would the prediction remain if the object appeared in a different environment?* To answer this at inference time, we estimate object and background expectations within CLIP's representation space, and synthesize counterfactual embeddings by recombining object features with diverse alternative contexts sampled from external datasets, batch neighbors, or text-derived descriptions. By estimating the Total Direct Effect and simulating intervention, we further subtract background-only activation, preserving beneficial object–context interactions while mitigating hallucinated scores. Without retraining or prompt design, our method substantially improves both worst-group and average accuracy on context-sensitive benchmarks, establishing a new zero-shot state of the art. Beyond performance, our framework provides a lightweight representation-level counterfactual approach, offering a practical causal avenue for debiased and reliable multimodal reasoning. The implementation is available at https://github.com/peipeng98.

## 1    Introduction

Vision language models, such as CLIP [1], have demonstrated impressive success by embedding images and textual descriptions into a shared semantic space, enabling strong zero-shot recognition across diverse datasets [2, 3, 4]. By training on large-scale natural image-text pairs, these models aim to generalize beyond finite training distributions. However, recent studies [5] highlight a critical limitation: Real-world datasets inherently encode strong co-occurrence patterns between target classes and contextual cues, including background scenes and their complex interactions. These spurious correlations provide shortcut signals during training [6, 7], leading models to rely on context rather than the true causal semantics, and ultimately degrading their zero-shot recognition performance, particularly when the test-time context diverges from familiar training distribution.

The issue is especially pronounced in CLIP, where training objectives that maximize mutual information across modalities unintentionally reinforce object-context entanglement. During optimization, frequent co-occurrence patterns amplify semantic signals from contextual features, causing background cues, co-occurring entities and scene-specific interactions to dominate the learned representations. This focus shift skews decision boundaries away from intrinsic object feature. As shown in Fig. 1(a), although such biases may enhance performance on in-distribution data, they substantially degrade model performance when confronted with novel backgrounds [8, 9, 10, 11].

In response to the contextual entanglement, particularly the visual hallucinations induced by background signals, recent strategies generally fall into two directions. A common and lightweight

---

\*Correspondence to Sheng-Jun Huang: huangsj@nuaa.edu.cn

39th Conference on Neural Information Processing Systems (NeurIPS 2025).

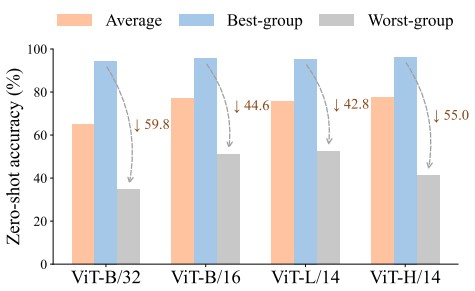

(a) Accuracy drop under co-occurrence bias

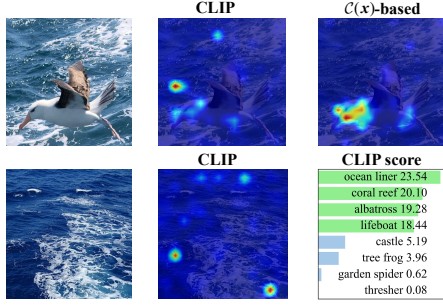

(b) Visual bias and model hallucinations

Figure 1: **Bias and hallucinations caused by co-occurrence in CLIP**. (a) Accuracy decreases significantly from the best to worst group across backbones in Waterbirds dataset, with groups defined by different class–context co-occurrence patterns. (b) The Attention maps show image responses under the prompt "a photo of an albatross," comparing CLIP and our counterfactual embedding $\mathcal{C}(\boldsymbol{x})$, and context-only images trigger hallucinated score on ImageNet labels, highlighting vision-side bias.

indirect solution is to augment the textual modality, typically by enriching prompts with additional contextual descriptors [12, 13, 14, 15]. However, two fundamental challenges remain: (i) generating comprehensive and accurate descriptions of the visual context is inherently challenging, and (ii) the text encoder itself inherits co-occurrence biases from pretraining, which may further entangle object and context semantics rather than disentangle them. On the other hand, directly operating on the visual modality provides a more principled alternative that avoids the challenges in textual prompt-based methods. Yet current approach [16, 17, 18, 19] often rely on generative models, complex augmentation pipelines, or fine-tuning of pretrained backbones, resulting in increased computational cost and limited scalability in practice. Detailed work is available in Appendix A.

Thus, motivated by causal inference [20, 21, 22], we propose a novel framework: instead of retraining models or textual augmentation, we enable model to imagine counterfactual scenarios directly at inference time and let it autonomously answer the causal question: "*Would this prediction remain consistent if the object appeared in a different environment?*" Specifically, we separately estimate the conditional expectations of object (target entity) and background (contextual cues), then construct their respective optimal estimation in the representation space. It will intervene objects by simulating in alternative contexts, derived from external scene datasets, internal scene from batch neighbors, or virtual scene descriptions constructed within the model's semantic space.

Additionally, to suppress hallucinated background effects while preserving the beneficial interaction effects captured by the model, we introduce an image-level Total Direct Effect (TDE) computation, which explicitly quantifies and removes the illusory influence arising from high-frequency object-context co-occurrences. Notably, our method requires no additional training, no external generative modules, and no handcrafted prompts, offering a lightweight, scalable, and causal solution for real-world deployment. In summary, our contributions are as follows:

- We demonstrate a causal perspective to model co-occurrence biases in terms of beneficial interactions and potential hallucinations, and explain the sensitivity of prompt-based strategies.

- We propose a counterfactual estimation method to operate directly at the representation level and develop a lightweight, inference-only debias scheme. It synthesizes counterfactual embeddings by leveraging skills (e.g. scene descriptions), and employs targeted interventions and TDE estimation to eliminate hallucinated effects while preserving beneficial object-context interactions.

- We present our method achieves state-of-the-art zero-shot recognition performance across multiple context-sensitive datasets, significantly improving model's reliability under distribution shifts.

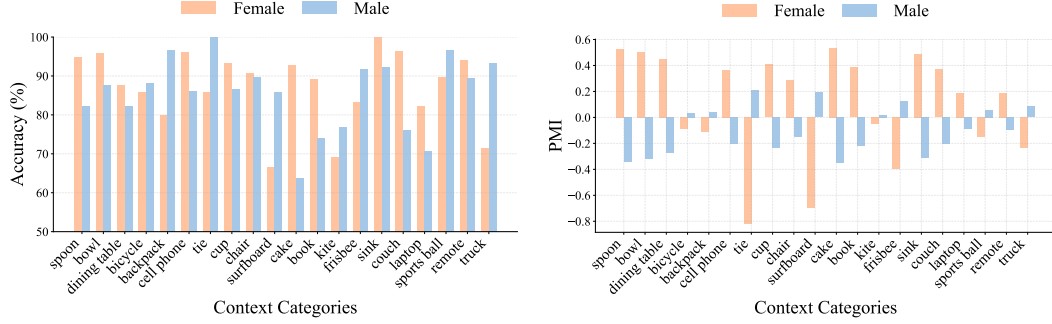

(a) Gender-specific accuracy on contextual objects

(b) PMI between gender and contextual objects

Figure 2: **Context-induced gender bias in CLIP.** (a) Zero-shot accuracy gap on COCO-GB v1 across genders and contextual objects. (b) Corresponding PMI revealing co-occurrence bias pattern.

## 2 Revisiting Object–Context Bias in CLIP

**Setup.** Let $X \in \mathcal{X}$ and $Z \in \mathcal{Z}$ denote, respectively, the object category (labeled as $Y \in \mathcal{Y}$) and contextual background.[2] Each image instance $\mathbf{i} \in \mathcal{I}$ is generated from $p(\mathbf{i} \mid X=\boldsymbol{x}, Z=\boldsymbol{z})$. CLIP transforms image via a visual encoder $f_i : \mathcal{I} \to \mathbb{R}^d$ and prompt $T$ via a textual encoder $f_t : \mathcal{T} \to \mathbb{R}^d$. For brevity, we let $\mathbb{E}[f_i \mid X=\boldsymbol{x}]$ abbreviate to $\mathbb{E}_{\mathbf{i} \sim p(\mathbf{i}|X=\boldsymbol{x})}[f_i(\mathbf{i})]$, and similarly for $Z = \boldsymbol{z}$.

**Two-Factor Decompositions in CLIP Modalities.** Following the Hoeffding perspective (a.k.a. ANOVA decompositions) [23, 24, 25], with respect to object–context factors, the instance $\mathbf{i}$ can be decomposed into the following structural causal function:

$$f_i(\mathbf{i}) = \boldsymbol{e}_i(\boldsymbol{x}) + \boldsymbol{e}_i(\boldsymbol{z}) + r_i(\boldsymbol{x}, \boldsymbol{z}) + \eta_i, \tag{1}$$

where $\boldsymbol{e}_i(\boldsymbol{x}) = \mathbb{E}_{\mathbf{i} \sim p(\mathbf{i}|X=\boldsymbol{x})}[f_i(\mathbf{i}) \mid X=\boldsymbol{x}]$ and $\boldsymbol{e}_i(\boldsymbol{z}) = \mathbb{E}_{\mathbf{i} \sim p(\mathbf{i}|Z=\boldsymbol{z})}[f_i(\mathbf{i}) \mid Z=\boldsymbol{z}]$ are the first–order main effects of the object and background, $r_i(\boldsymbol{x}, \boldsymbol{z})$ is the interaction effects, and $\eta_i$ summarizes the higher-order residual. It is proved in Appendix. B.1.1 that this is the only pairwise orthogonal expansion of these terms of Eq. (1).

Similarly, the composite prompt $T = (T_x, T_z)$ is embedded as

$$f_t(T) = \boldsymbol{e}_t(\boldsymbol{x}) + \boldsymbol{e}_t(\boldsymbol{z}) + r_t(\boldsymbol{x}, \boldsymbol{z}) + \eta_t, \tag{2}$$

where $\boldsymbol{e}_i(\boldsymbol{x})$ and $\boldsymbol{e}_i(\boldsymbol{z})$ are the textual modal main effects; $r_t(\boldsymbol{x}, \boldsymbol{z})$ and $\eta_t$ are respectively interaction terms and residual. Detailed derivations of Eq. (2) appear in Appendix B.1.2.

**Co-occurrence Bias Emerges from Contrastive Learning.** The InfoNCE objective used in CLIP maximizes the alignment between paired image and text embeddings while pushing apart mismatched ones. When representations are decomposed into object and context components, the InfoNCE gradient reveals a distinct structure: for any object–scene pair $(\boldsymbol{x}, \boldsymbol{z})$, high co-occurrence frequency leads to large gradient contributions on amplifying the interaction terms $r_i, r_t$, and the cross-modal similarity of $\langle \boldsymbol{e}_i(\boldsymbol{z}), \boldsymbol{e}_t(\boldsymbol{x}) \rangle$, proved in Appendix B.2 and B.3. These components widen the margin between positive samples and their single-factor negatives (e.g., $(\boldsymbol{x}, \boldsymbol{z}')$ or $(\boldsymbol{x}', \boldsymbol{z})$).

Thus, the biases are enhanced during training and governed by the statistical co-occurrence of $(\boldsymbol{x}, \boldsymbol{z})$, which can be quantified by the Pointwise Mutual Information (PMI):

$$\text{PMI}(\boldsymbol{x}, \boldsymbol{z}) = \log \frac{p(\boldsymbol{x}, \boldsymbol{z})}{p(\boldsymbol{x}) \, p(\boldsymbol{z})}, \tag{3}$$

which measures the extent to which a pair of variables co-occurs more frequently than expected under statistically independent. Since mutual information $I(X; Z)$ is the expectation of PMI over $(\boldsymbol{x}, \boldsymbol{z})$, and the InfoNCE loss is a lower bound on $I(X, Z; T)$, minimizing InfoNCE inherently promotes high PMI pairs in cross-modal alignment. The detail are shown in Appendix B.4.

---

[2]Both label sets are finite, with $Z$ enumerating all scenes present in the dataset.

As a result, unless a dataset samples objects and contexts uniformly, minimizing the InfoNCE loss will inherently entangle them, amplifying both interaction energy and cross-modal similarity. This dependency is difficult to avoid in practice and is a key source of context-induced misclassifications in vision–language models. Fig. 2 provides a empirical evidence: on a gender-balanced subset COCO-GB v1 [10], the zero-shot accuracy gap between female and male correlates with their PMI to context, revealing similar gender–context co-occurrence biases shared between COCO and LAION-2B [26]. These results confirm that CLIP models internalize context-driven associations, which persist even under zero-shot settings and impact fair prediction across different groups.

**Causal Explanation for Hallucination-Induced Misclassification.** At inference, the relationship of causal effect can be constructed as shown in Fig. 3, where node $\mathcal{I}$ feeds two children: object $X$ and scene $Z$; a solid arrow $X \rightarrow Y$ represents the *ideal* decision path, whereas the dashed links between $Z$ and $X$ indicate statistical co-occurrence learned during contrastive training. When conditioning on $\mathcal{I}$, extra predictive paths emerge, including $Z \rightarrow Y$ and $(X, Z) \rightarrow R \rightarrow Y$. Thus, the CLIP score[3] for any class $c \in \mathcal{Y}$ with an object-only prompt expands to

$$S(\mathbf{i}, T_c) = \langle \boldsymbol{e}_i(\boldsymbol{x}), f_t(T_c) \rangle + \langle \boldsymbol{e}_i(\boldsymbol{z}), f_t(T_c) \rangle + \langle \boldsymbol{r}_i(\boldsymbol{x}, \boldsymbol{z}), f_t(T_c) \rangle + \langle \eta_i, f_t(T_c) \rangle. \quad (4)$$

The inner product $\langle \boldsymbol{e}_i(\boldsymbol{z}), f_t(T_c) \rangle$ is the pure context score. It remains non-zero even for "background-only" images due to image encoder bias (see Fig. 1(b)), and may thus produce *hallucination* when the score favors a class in which the image does not exist. The term $\langle \boldsymbol{e}_i(\boldsymbol{x}), f_t(T_{c_x}) \rangle$ is the ideal target, and the interaction term $\langle \boldsymbol{r}_i(\boldsymbol{x}, \boldsymbol{z}), f_t(T_{c_x}) \rangle$ acts as an object-specific prior when $(\boldsymbol{x}, \boldsymbol{z})$ really co-occur, but drops to low-level for totally novel pairs.

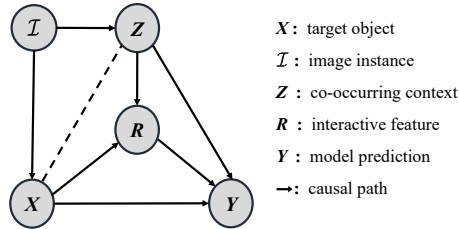

$X$ : target object
$\mathcal{I}$ : image instance
$Z$ : co-occurring context
$R$ : interactive feature
$Y$ : model prediction
$\rightarrow$ : causal path

Figure 3: **The schematic of causal graph.** Dashed line denotes $X$ and $Z$ co-occurring in the same image. $R$ represents the interaction as observable features.

When $(\boldsymbol{x}, \boldsymbol{z})$ follows the training distribution, the ideal target dominates the CLIP score, and the scene score and $r_i$ reinforce correct decision boundaries, just as contrastive training dose. However, for a novel pairing $(\boldsymbol{x}, \boldsymbol{z}')$ which rarely or never co-occurred during training, the interaction $r_i$ remains a low level. Meanwhile the frequent scene $z'$ favors the class $c_{x'}$ that often co-occurred with it during training stage. The decision margin between the correct class $c_x$ and the confusable class $c_{x'}$ is

$$\begin{aligned} \delta &= S(\mathbf{i}_{x,z}, T_{c_x}) - S(\mathbf{i}_{x,z}, T_{c_{x'}}) \\ &\approx \left( \langle \boldsymbol{e}_i(\boldsymbol{x}), f_t(T_{c_x}) \rangle - \langle \boldsymbol{e}_i(\boldsymbol{x}), f_t(T_{c_{x'}}) \rangle \right) - \left( \langle \boldsymbol{e}_i(\boldsymbol{z}'), f_t(T_{c_{x'}}) \rangle - \langle \boldsymbol{e}_i(\boldsymbol{z}'), f_t(T_{c_x}) \rangle \right), \end{aligned} \quad (5)$$

If the contextual effect inherited from co-occurrence pairs $(x', z')$ dominates target evidence, then $\delta < 0$ and prediction incorrectly favors $c_{x'}$, shown by zero-shot recognition failures cases in Fig. 1(a).

**Causal Explanation of Prompt-Based Context Intervention.** With the object-only prompt, the prediction implicitly marginalizes over all possible contexts, as a priors in encoder, which are derived from the training set and influenced by co-occurrence patterns, that is $p(Y \mid \mathcal{I}, T_{c_x}) = \sum_{\mathcal{Z}} p(Y \mid \mathcal{I}, T_{c_x}, Z = \boldsymbol{z}) \, p(Z = \boldsymbol{z} \mid \mathcal{I})$, so the shortcut $\mathcal{I} \rightarrow Z \rightarrow Y$ remains active. Appending an explicit scene token $T_z$ amounts to conditioning on (equivalently intervening in) $Z$:

$$p\big(Y \mid \mathcal{I}, T_{c_x}, T_z\big) = p\big(Y \mid \mathcal{I}, T_{c_x}, Z = \boldsymbol{z}\big), \quad (6)$$

thus clamping the latent context and cutting off the spurious branch $\mathcal{I} \rightarrow Z \rightarrow Y$. The classifier then relies solely on the legitimate $\mathcal{I} \rightarrow X \rightarrow Y$ pathway. But these methods [12, 13, 14, 15] have to use an accurate $T_z$ neutralizes the $Z \rightarrow Y$ shortcut. Imprecise or mismatched $T_{z'}$ only partially cancels the bias and may divert information to unrelated classes, even bringing more bias by $T_z$, which is accounting for the empirical sensitivity of prompt engineering. Details are shown in Appendix B.5.

**Towards a Counterfactual Solution.** Causal inference provides a principled approach to mitigating spurious correlations by identifying and suppressing causally irrelevant signals. However, existing methods often rely on generative models [27, 28], auxiliary networks [29, 30], or counterfactual losses terms [31], which increase model complexity and limit applicability in zero-shot setting. Considering these limitations, as shown in Fig. 4, we propose a lightweight, inference-only framework

---

[3]In practice, the CLIP score is normalized, and we simplify the derivation by omitting the scaling factor $1/\tau$.

that intervenes directly within the CLIP representation space. Inspired by the principle of randomized experiments, our method synthesizes counterfactual embeddings by recombining the object with diverse, label-agnostic scene contexts, thus suppressing background-induced illusions while preserving useful object–context interactions. This allows for causal debiasing without modifying the model architecture or relying on handcrafted prompts.

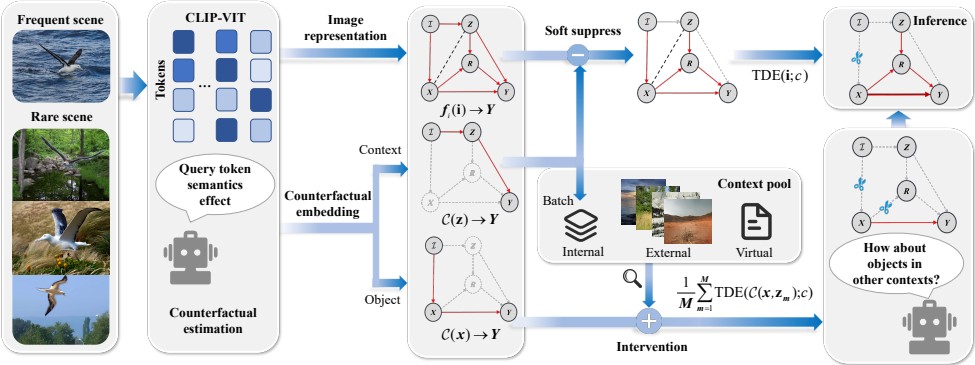

Figure 4: **Overall architecture**. Our method consists of two components: the upper branch computes TDE by subtracting the predictive contribution of context; the lower branch constructs counterfactual embeddings by recombining object with diverse alternative contexts embeddings, simulating intervention. Aggregating those yields a robust, decoupled prediction. The gray dashed line, and red arrows denote blocked, preserved casual effect to prediction respectively. A detailed step-by-step description of our inference pipeline can be found in Algorithm 1 in Appendix E.4.

# 3 Counterfactual Inference at Representation Level for Zero-shot Learning

## 3.1 Estimating Counterfactual at Representational Level

A basic challenge in counterfactual reasoning lies in its numerical computation. From a novel angle, we estimate the counterfactual embedding directly from the original image without any extra model.

**Token-Level Direct-Effect Decomposition.** We focus on ViT [32] as the backbone of CLIP, built from $L$ layers, each of which contains a Multi-Head self-Attention (MSA) followed by an MLP block. The representation $f_i(\mathbf{i})$ is a linear projection $P \in \mathbb{R}^{d' \times d}$ of ViT output to joint vision-and-language space. The matrix $\mathbf{t}^0 \in \mathbb{R}^{d \times (N+1)}$, with tokens $t_{\text{cls}}^0, t_1^0, \ldots, t_N^0$ as columns, constitutes the initial state of residual stream. [4] Following the residual updates, $f_i(\mathbf{i})$ is split into

$$f_i(\mathbf{i}) = P\, t_{\text{cls}}^0 + \sum_{l=1}^{L} P\big[\text{MSA}^l(\text{LN}_1^l(\mathbf{t}^{l-1}))\big]_{\text{cls}} + \sum_{l=1}^{L} P\big[\text{MLP}^l(\text{LN}_2^l(\hat{\mathbf{t}}^l))\big]_{\text{cls}}, \tag{7}$$

where pre–projection LayerNorm $\text{LN}(\cdot)$ can be absorbed into $P$ and its bias is evenly spread across all summations. Thus, the image embedding are divided into three direct-effect terms of initial class token, MSAs and MLPs. Following previous work [33, 18], MSA output rewrites a sum over $H$ attention heads and $N$ tokens as

$$P\big[\text{MSA}^l(\text{LN}_1^l(\mathbf{t}^{l-1}))\big]_{\text{cls}} = \sum_{h=1}^{H} \sum_{j=0}^{N} u_j^{l,h}, \quad u_j^{l,h} = \alpha_j^{l,h}\, W_{VO}^{l,h}\text{LN}_1^l(t_j^{l-1}), \tag{8}$$

where $W_{VO}^{l,h} \in \mathbb{R}^{d \times d}$ are transition matrices and $\alpha_j^{l,h} \in \mathbb{R}$ are the attention weights from class token to the $j$-th token, which satisfy $\sum_{j=0}^{N} \alpha_j^{l,h} = 1$. More algebraic proofs are provided in Appendix B.6.

To investigate the effect of the component, we used average ablation [34], replacing the component with its average over batch, for measuring the decrease in accuracy after ablation. Experiments on

---

[4]The exact form $\hat{\mathbf{t}}^l = \text{MSA}^l(\text{LN}_1^l(\mathbf{t}^{l-1})) + \mathbf{t}^{l-1}, \mathbf{t}^l = \text{MLP}^l(\text{LN}_2^l(\hat{\mathbf{t}}^l)) + \hat{\mathbf{t}}^l.$

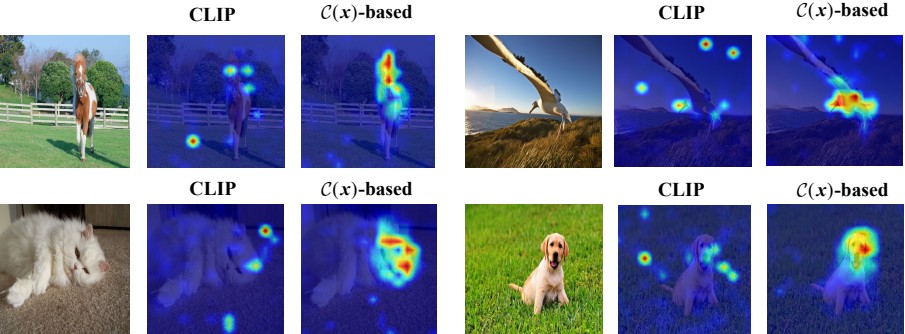

Figure 5: **The attention map revealed by $\mathcal{C}(x)$-based embeddings on NICO.**

ImageNet with multiple backbones in Tab. 5 found that the accuracy changed to less than $0.1\%$ after MSA ablation, whereas after ablation for the effect of initial class token and MLP, it only decreased by about $1\%$ compared to the baseline. Therefore, it is reasonable to design the main effect of each token $t_j^0$ on $f_i(\mathbf{i})$ as a direct effect of the MSAs and a bias terms on the initial class token and MLPs after ablation. Hence, we define the semantic token effect:

$$v_j(\mathbf{i}) = \sum_{l=1}^{L} \sum_{h=1}^{H} u_j^{l,h} + \frac{1}{N}\varepsilon, \tag{9}$$

where $\varepsilon$ is constant for batch about class token and MLPs after mean ablation.

**Optimal First-Order Estimates for Counterfactual Embeddings.** Let an image $\mathbf{i}$ be partitioned into $N$ patch-tokens $\{t_1, t_2, \ldots, t_N\}$, and their semantic effect obtain from Eq. (9). We define the counterfactual embedding as $\mathcal{C}(\boldsymbol{x}, \boldsymbol{z}')$. Specifically, when $\boldsymbol{z}'$ is set to base state, [5] it simplifies to $\mathcal{C}(\boldsymbol{x})$, which contains only the embedding of $\boldsymbol{x}$. Analogously, $\mathcal{C}(\boldsymbol{z})$ is defined in the same way.

We introduce a Bernoulli latent variable $G$ indicate semantic property of image $\mathbf{i}$ whether it is context $\boldsymbol{z}$ or target object $\boldsymbol{x}$. Its corresponding conditional distribution on instance $\mathbf{i}$ is defined as $p(f_i(\mathbf{i}) \mid G=\boldsymbol{z})$, which means when we focus on background parts, how does embedding distribution look like. This exactly corresponds to $\mathcal{C}(\boldsymbol{z})$, representing the case there is only the background without any target objects. Formally, define $\mathcal{C}(\boldsymbol{z})$ as expectation vector under this condition:

$$\mathcal{C}(\boldsymbol{z}) = \mathbb{E}[f_i(\mathbf{i}) \mid G = \boldsymbol{z}] = \int \tau p(\tau \mid G = \boldsymbol{z})d\tau, \tag{10}$$

It is the center of gravity in the embedding space when "looking only at the background", and the optimal first-order estimate for the background counterfactual embedding in the image. Detailed proofs are provided in the Appendix B.7. And from visualization in Fig. 5, it confirms that $\mathcal{C}(x)$ can enable better attentional focus on target objects.

In practice, we discretize Eq. (10). Assuming image is sampled uniformly and that all token priors are equal, Bayes' theorem gives $p(v_j(\mathbf{i}) \mid G = \boldsymbol{z}) \propto p(G = \boldsymbol{z} \mid v_j(\mathbf{i}))$. Therefore, the counterfactual embedding $\mathcal{C}(\boldsymbol{z})$ can be computed as:

$$\mathcal{C}(\boldsymbol{z}) \approx \sum_{j}^{N} v_j(\mathbf{i})p(t_j \mid G = \boldsymbol{z}) = \frac{\sum_{j}^{N} w_z(t_j)v_j(\mathbf{i})}{\sum_{j}^{N} w_z(t_j)}, \quad w_z(t_j) = p(G = \boldsymbol{z} \mid v_j(\mathbf{i})), \tag{11}$$

where normalizes the weights $w_z(t_j)$ to ensure the results in the expectation form. And, to stay aligned with the rest of the CLIP embedding (which are all in the unit sphere), we usually do another $L_2$ normalization for $\mathcal{C}(\boldsymbol{z})$. $\mathcal{C}(\boldsymbol{x})$ can obtain in the same way by $w_x$ and $v_j(\mathbf{i})$.

Then, as for each token $t_j$ and class $c_x$, we compute its sigmoid score:

$$w_x(t_j) = p(G = c_x \mid v_j(\mathbf{i})) = \sigma(S(t_j \mid T_{c_x})), \tag{12}$$

---

[5] The base state sets the variable's effect to a fixed constant, defaulting to zero in this paper.

where $\sigma$ is sigmoid function, $S$ is the CLIP score mentioned before. We prefer sigmoid (over softmax) because it treats each class one-vs-rest, yielding independent probabilities and avoiding the need for any hand-crafted "background" prompt. We thus define the token's background probability by

$$w_z(t_j) = p(G = \boldsymbol{z} \mid v_j(\mathbf{i})) = 1 - \max_{c \in Y} \sigma(S(t_j \mid T_c)), \tag{13}$$

This mutual-exclusion rule treats background as the complement of the most likely foreground class, which empirically provides a sharp, unambiguous estimate of $w_z$.

## 3.2  TDE-Driven Context Preservation and Hallucination Suppression

Let $\mathbf{i}_{x,z}$ contains object $\boldsymbol{x}$ and context $\boldsymbol{z}$, as for any class $\boldsymbol{c} \in \mathcal{Y}$, conventional reasoning is for computing the Total Effect: $\text{TE}(\mathbf{i}_{x,z}; c) = p(Y{=}c \mid \mathbf{i}) - p(Y{=}c \mid \mathbf{i}_{x_0,z_0})$, where $\mathbf{i}_{x_0,z_0}$ means base state image. As shown in Fig. 3, TE-based predictions are influenced by both benign interaction effects and malignant hallucination effects. Total Direct Effect (TDE) provides the most straightforward approach: preserving useful pathways $(X, Z) \to R \to Y$ and suppressing hallucination channel $Z \to Y$, we obtain:

$$\text{TDE}(\mathbf{i}; c) = p(Y = c \mid \mathbf{i}_{x,z}) - p(Y = c \mid \mathbf{i}_{x_0,z}) = p(Y = c \mid \mathbf{i}_{x,z}) - p(Y = c \mid \mathcal{C}(\boldsymbol{x}_0, \boldsymbol{z})), \tag{14}$$

Obviously, counterfactual embedding via Eq. (11) can isolate malignant hallucinatory effect. In CLIP, the posterior takes the form $p(Y{=}c \mid \mathbf{i}){=}\sigma(S(\mathbf{i}, T_c))$. We thus update the TDE in logit space as

$$\text{TDE}(\mathbf{i}; c) = S(\mathbf{i}_{x,z}, T_c) - \hat{\lambda}S(\mathcal{C}(\boldsymbol{z}), T_c), \tag{15}$$

where $\hat{\lambda}$ is a suppression coefficient controlling the contribution of background-only signals. Because of the strictly increase of $\sigma(\cdot)$, the form of Eq. (15) satisfies order preservation and avoids the numerical saturation of sigmoid. Therefore, the logit-based TDE is used throughout our experiments to isolate beneficial object–scene cues and suppress hallucination.

## 3.3  Effortless Counterfactual Construction and Intervention on Inference

From the other side, we tent to simulate intervention that apply the do-operator on $X$ for eradicating the path $X \leftarrow \mathcal{I} \to Y$, leaving only the genuine object–label causal edge $X{\to}Y$. Unlike the soft mitigation approach in the previous section, it presents a direct solution to yield a more purified ideal effect $X \to Y$. To be exact, we select a set of optimal context embeddings to synthesize counterfactual image embeddings, which serve as classifier inputs to simulate intervention. As the causal graph Fig. 3 satisfies the back-door criterion [20, 35], we define:

$$p(Y = c \mid do(\boldsymbol{x})) = \sum_{m=1}^{M} p(Y = c \mid X = \boldsymbol{x}, Z = \boldsymbol{z}_m)p(Z = \boldsymbol{z}_m), \tag{16}$$

where $M$ denotes counterfactual sample size, and do-operator [20, 35] formally models intervention.

**Sources of Candidate Context Embedding.** More comprehensive candidate context embeddings yield enhanced intervention disentanglement. We consider three pools of context embeddings $\{f_i(\boldsymbol{z}_b)\}_{b=1}^{B}$ corresponding to three experimental setups. (I) **External scene** datasets (e.g. Places-365 [36]): It samples dataset directly by CLIP image encoder to get $f_i(\boldsymbol{z})$. (II) **Internal scene** from batch: It samples counterfactual context embedding $\mathcal{C}(\boldsymbol{z}_b)$ in a batch by Eq. (11). (III) **Virtual scene by description**: Leveraging CLIP's shared semantic space enables effortless generation of scene descriptions $f_t(T_{z_b})$ to serve as virtual counterfactual context embeddings.

For context embeddings obtained above, a filter sampler is designed to improve its quality. It evaluates each candidate embedding for image $\mathbf{i}_{x,y}$, using scoring function based on cosine similarities between $f_i(\boldsymbol{z}_b)$ and $\mathcal{C}(\boldsymbol{x})$ as well as $f_i(\boldsymbol{z}_b)$ and $\mathcal{C}(\boldsymbol{z})$, preferentially selecting embeddings with lower combined scores to maximize scene diversity. Thus, define the synthesized counterfactual embeddings [6] as

$$\mathcal{C}(\boldsymbol{x}, \boldsymbol{z}_m) = \alpha\mathcal{C}(\boldsymbol{x}) + (1 - \alpha)f_i(\boldsymbol{z}_m), \quad \alpha \in (0, 1), \tag{17}$$

Then, assuming a uniform sampling and considering the new bias from $f_i(\boldsymbol{z})$, Eq. (16) turns into

$$\frac{1}{M}\sum_{m=1}^{M}\text{TDE}(\mathcal{C}(\boldsymbol{x}, \boldsymbol{z}_m); c) = \frac{1}{M}\sum_{m=1}^{M}\big(S(C(\boldsymbol{x}, \boldsymbol{z}_m), T_c) - \hat{\lambda}S(f_i(\boldsymbol{z}_m), T_c)\big), \tag{18}$$

---

[6]The synthesized embedding will be $L_2$ normalized by default.

This form can effectively prevent potential bias about $Z_m \to Y$ in introduced scenes. Combining Eq. (15) and (18), we obtain

$$arg \max_{c \in Y} y(\mathbf{i}_{x,z}; c), \quad y(\mathbf{i}_{x,z}; c) = (1 - \lambda)\text{TDE}(\mathbf{i}; c) + \lambda \frac{1}{M} \sum_{m=1}^{M} \text{TDE}(\mathcal{C}(\boldsymbol{x}, \boldsymbol{z}_m); c), \quad (19)$$

where $\lambda$ controls for the extent to which interactions in the original image $\mathbf{i}$ affect prediction. From another angle, Eq. (19) is interpreted as the model's imagination beyond image $\mathbf{i}$. By conceptualizing different scenarios $\boldsymbol{z}_m$, combining the original scene $\boldsymbol{z}$ through weights, a more robust and fair prediction result can be obtained.

## 4  Experiment

**Settings.** We evaluate our method on four widely adopted benchmarks that target context-sensitive distribution shifts: Waterbirds [8], UrbanCars [9], COCO-GB [10], and NICO [11]. These datasets span various real-world correlations, including different out-of-distribution context. We report average and worst-group accuracy to assess both overall performance and vulnerability to specific context–label combinations under zero-shot classification. Detailed dataset information, implementation details, and variant configurations are provided in Appendix C.

**Performance on Waterbirds.** In Tab. 1, Our methods consistently achieve the SOTA in both average and worst-group accuracy across all backbones. Compared to vanilla CLIP, it improves worst-group accuracy by an average of $+32.3\%$, with a remarkable gain of $+44.6\%$ on ViT-B/32. Meanwhile, average accuracy also sees significant improvement, with an average increase of $+14.3\%$, and a maximum gain of $+19.1\%$ on ViT-B/32. In contrast, other methods such as PC$^+$ introduces textual bias (see Fig. 8) when limited fine-grained prompt is used, trading off average accuracy to improve worst-group performance. Detailed results about Waterbirds are shown in Appendix D.1.

Table 1: **Average and worst-group accuracy (%) on Waterbirds on CLIP backbones.** Best results are in **bold**, top-2 in *gray*. PC$^+$ selects limited fine-grained prompts than coarse-grained PC.

| Method | ViT-B/32 | | ViT-B/16 | | ViT-L/14 | | ViT-H/14 | |
|---|---|---|---|---|---|---|---|---|
| | Avg.↑ | Worst↑ | Avg.↑ | Worst↑ | Avg.↑ | Worst↑ | Avg.↑ | Worst↑ |
| CLIP (CVPR$'$21 [1]) | 64.88 | 34.55 | 77.20 | 50.93 | 75.65 | 52.51 | 77.55 | 41.28 |
| TBD (ICLR$'$24 [18]) | 66.98 | 38.63 | 83.88 | 64.17 | 85.04 | 75.25 | 84.98 | 48.91 |
| PC (ICLR$'$24 [15]) | 71.73 | 57.25 | 81.79 | 55.45 | 87.83 | 70.72 | 82.05 | 45.02 |
| PC$^+$ (ICLR$'$24 [15]) | 65.27 | 46.12 | 79.06 | 66.67 | 81.65 | 75.08 | 76.35 | 66.82 |
| B2T (CVPR$'$24 [14]) | 68.59 | 56.23 | 77.46 | 64.64 | 84.12 | 48.13 | 77.25 | 46.57 |
| Ours (External) | 79.96 | **79.16** | 86.40 | 70.87 | 91.08 | 82.09 | 88.23 | 63.40 |
| Ours (Internal) | 81.93 | 71.81 | 85.43 | **82.71** | 90.71 | **85.67** | 87.66 | 67.45 |
| Ours (Virtual) | **83.69** | 65.73 | **89.47** | 67.13 | **91.94** | 80.84 | **88.70** | **68.38** |

Table 2: **Performance on COCO-GB v1 on ViT-B/16.** We report average, female, and male subgroup accuracies, as well as the gender performance gap. And a smaller gap indicates better fairness across gender groups.

| Method | Avg.↑ | Female↑ | Male↑ | Gap↓ |
|---|---|---|---|---|
| CLIP | 88.70 | 85.60 | 91.80 | 6.20 |
| TBD | 90.80 | 89.60 | 92.00 | 2.40 |
| PC | 90.50 | 89.00 | 92.00 | 3.00 |
| PC$^+$ | 90.40 | 88.20 | **92.60** | 4.40 |
| B2T | 90.10 | 89.40 | 90.80 | 1.40 |
| Ours (external) | 91.10 | 89.80 | 92.40 | 2.60 |
| Ours (internal) | 91.25 | 90.50 | 92.00 | 1.50 |
| Ours (virtual) | **91.40** | **91.80** | 91.00 | **0.80** |

Table 3: **Performance on UrbanCars on ViT-B/16.** We report accuracy on the original co-occurrence (I.D.), background shifted (BG), and Co-object shifted (Co-Obj) subsets, along with overall average accuracy.

| Method | Avg.↑ | I.D.↑ | BG↑ | Co-Obj↑ |
|---|---|---|---|---|
| CLIP | 63.07 | 82.00 | 37.20 | 70.00 |
| TBD | 53.87 | 58.00 | **50.80** | 52.80 |
| PC | 64.67 | 79.60 | 46.40 | 68.00 |
| PC$^+$ | 65.33 | 83.60 | 43.20 | 69.20 |
| B2T | 63.40 | 78.40 | 44.40 | 67.40 |
| Ours(External) | 68.53 | 86.40 | 45.60 | 73.60 |
| Ours(Internal) | 68.27 | 88.00 | 45.60 | 71.20 |
| Ours(Virtual) | **71.87** | **89.60** | 48.00 | **78.00** |

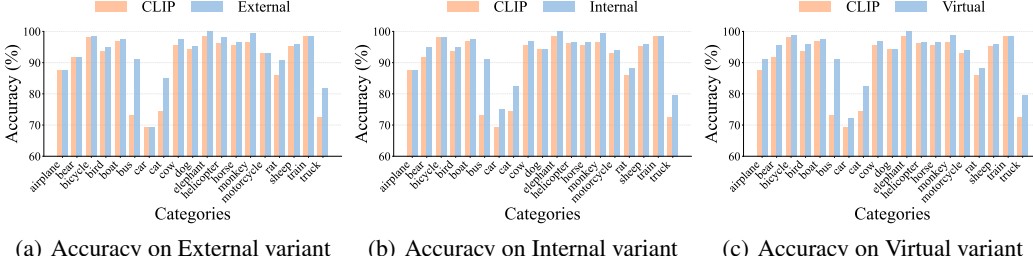

(a) Accuracy on External variant  (b) Accuracy on Internal variant  (c) Accuracy on Virtual variant

Figure 6: **Per-Class worst-group accuracy with CLIP (ViT-B/16) on NICO.** Comparison of worst-group accuracy across 19 categories on NICO (ViT-B/16) using CLIP and our external, internal, and virtual variants. Our method consistently improves the lowest-performing context for each class.

**Performance on COCO-GB and UrbanCars.** Tab. 2 presents results of a subset about gender on the MS-COCO, demonstrating that although CLIP is pre-trained on large-scale real-world data, it remains vulnerable to contextual co-occurrence bias, particularly evident in gender prediction (see Fig. 2). For example, CLIP exhibits a notable performance gap of $6.20\%$ between male and female samples. While PC$^+$ improves male accuracy, it further amplifies gender disparity. Tab. 3 further evaluates performance on UrbanCars. By analyzing samples with background shifts and co-occurring object shifts, we observe substantial accuracy drops for CLIP, revealing its reliance on spurious, non-discriminative features. Although TBD enhances robustness against background bias, it significantly deteriorates performance on identity-preserving samples (I.D.) due to architectural disruption. In contrast, our method maintains high average accuracy while consistently improving model generalization and stability. Additional results can be found in Appendix D.2 and D.3.

**Performance on NICO.** Fig. 6 shows performance improvements of our three variants over CLIP (ViT-B/16), demonstrating consistent gains across diverse class within context-shifted dataset. The detailed comparisons are shown in Appendix D.4. Besides, we compare attention maps [37] of the counterfactual $\mathcal{C}_x$, with CLIP in NICO, and the Fig. 5 proves that the counterfactual module captures more complete embeddings about target object. Details are shown in Appendix D.4.

Table 4: **Ablation study on Waterbirds**. Average zero-shot accuracy (%) under different module configurations. ✗ indicates the ablation of the component: the **TDE** module in Eq. (15), **sampler** and **intervention** module via counterfactual calibration in Eq. (18).

| Method | Variant | Sampler | TDE | Intervention | ViT-B/32 | ViT-B/16 | ViT-L/14 | ViT-H/14 |
|---|---|---|---|---|---|---|---|---|
| CLIP | – | – | – | – | 64.88 | 77.20 | 75.65 | 77.55 |
| **Ours** | Random | | | | 65.19 | 59.94 | 77.36 | 69.55 |
| | External | | | ✗ | 75.70 | 79.57 | 87.26 | 84.21 |
| | | | ✗ | | 76.10 | 83.85 | 89.96 | 85.38 |
| | | ✗ | | | 79.92 | 84.97 | 90.89 | 87.50 |
| | | | | | **79.96** | **86.40** | **91.08** | **88.23** |
| | Internal | | | ✗ | 75.46 | 79.38 | 84.09 | 84.21 |
| | | | ✗ | | 77.94 | 82.29 | 87.80 | 82.10 |
| | | ✗ | | | 80.79 | 85.33 | 88.90 | 87.57 |
| | | | | | **81.93** | **85.43** | **90.71** | **87.66** |
| | Virtual | | | ✗ | 75.39 | 79.03 | 86.71 | 84.28 |
| | | | ✗ | | 82.21 | 88.47 | 88.16 | 86.38 |
| | | ✗ | | | 83.66 | 88.18 | 89.51 | 87.69 |
| | | | | | **83.69** | **89.47** | **91.94** | **88.70** |

**Ablation.** To assess the contribution of each module, we conduct ablations on three components. In Tab. 4, removing any component consistently degrades performance, and the full model achieves the highest gains, it confirms that these module are synergistic in mitigating contextual bias. Moreover, using sampler-filtered random vectors to ablate context pool confirms semantically specific contexts are required for counterfactual synthesis, while still outperforming the baseline in certain backbones.

More parameter setting and analysis are shown in Appendix C.2 and E.3. In addition, Appendix E.1 provides an extended comparison with multiple augmentation based baselines (e.g., Mixup, CutMix, AugMix, Cutout, and ALIA [28]), showing our method surpasses these zero-shot variants in accuracy.

**Efficiency.** Our method achieves significant improvements with remarkably low computational cost. A key advantage lies in performing counterfactual reasoning entirely at the representation level, which incurs much lower cost compared to conventional data augmentation methods. This design makes our approach highly scalable for large-scale datasets and high-throughput inference. For instance, on a single GPU, we can generate 100,000 counterfactual embeddings for a batch of 1,000 samples in less than 3 seconds, with effortless context embedding construction, particularly for the virtual variant.

To quantify efficiency further, Appendix E.2 provides detailed FLOPs analysis per image, assuming pre-computed text embeddings. Compared to diffusion-based ALIA and common augmentation methods, our representation-level variants incur only marginal additional cost ($+0.002 \sim 0.004$ G), while consistently improving accuracy across all backbones. In contrast, pixel-level approaches require several orders of magnitude more computation ($+10^3 \sim 10^6$ G), demonstrating the scalability and efficiency of our approach.

## 5   Conclusion

We propose a lightweight, inference-only framework that mitigates contextual hallucinations in vision–language models via representation-level causal reasoning. By estimating image embeddings into object and background components, we subtract spurious effects by the Total Direct Effect (TDE) measure, and simulate intervention by constructing counterfactual embeddings via recombining object features with diverse alternative contexts from external data, batch neighbors, and textual descriptions. Without requiring retraining, prompt tuning, or generative models, ours method achieves strong zero-shot performance and sets a new state-of-the-art on context-sensitive benchmarks.

## Acknowledgements

This work was supported by the NSFC(U2441285, 62222605).

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

# A   Extended Related Work

**Co-occurrence Bias in Vision–Language Models.** Vision–language models (VLMs) often learn spurious correlations from frequent concept co-occurrences in training data [7, 38], leading them to rely on context cues or stereotypes as shortcuts [5, 39]. For example, if men frequently appear repairing appliances, the model may incorrectly associate the activity with male gender [40]. Such biases degrade performance on unbiased cases and raise fairness concerns. Recent work shows that popular zero-shot VLMs exhibit notable gender disparities in retrieval and classification tasks [19], largely due to underrepresented attributes like gender, ethnicity, or background context. These issues highlight the need for effective debiasing strategies.

**Prompt-Based Bias Mitigation Techniques.** Prompt engineering mitigates bias in vision–language models by injecting context-aware cues into textual inputs. Some methods learn prompt embeddings adversarially to suppress spurious features such as gender or stereotypes [12], while others rewrite biased captions with more balanced phrasing [13]. These techniques are lightweight and compatible with pre-trained models, requiring no additional image data or architectural changes. However, their effectiveness depends heavily on task-specific keywords or priors. Poorly designed prompts may reduce performance or introduce new biases. To address this, recent work explores generating high-quality prompts, such as estimating contextual probabilities from images [15] or deriving targeted keywords through misclassification analysis [14].

**Data Augmentation and Model-level Strategies.** Data-centric methods mitigate bias by enriching the training distribution with additional image–text pairs that disrupt spurious co-occurrences. Concept graph-guided augmentation [16] targets rare object–context pairs but depends on generative models, raising concerns about output quality and scalability.

At the model level, contrastive approaches like CNC [17] align same-class samples across biases while separating spurious correlations, improving feature discrimination. Other methods prune biased classification heads [18] to reduce reliance on shortcut signals, though this may impair generalization. MEMO [41] adapt model predictions across multiple augmented views of each input by minimizing the entropy of their aggregated output distribution, thereby encouraging consistency under distribution shifts. While effective, these strategies typically require retraining or external modules, limiting their use in zero-shot settings.

**Causal and Counterfactual Reasoning Approaches.** Causal inference offers a principled way to mitigate spurious correlations by modeling the data-generating process. Unlike prompt tuning or data augmentation, causal methods operate directly on variables to estimate and suppress undesired biases. In vision–language tasks, counterfactual reasoning has been used to reweight or subtract biased modality signals, such as in counterfactual VQA [42], which isolates language priors using Total Direct Effect (TDE).

Recent methods extend this idea by decomposing representations via generative models [27], training dual-branch architectures to decouple confounders [29], or applying TDE at the patch level to weaken local shortcuts [31]. However, these approaches often require segmentation, auxiliary branches, or retraining, limiting their practicality in zero-shot or inference-only settings.

In contrast, representation-level counterfactual methods are well-suitable to zero-shot VLMs due to their simplicity and compatibility. Combining causal reasoning with efficient inference-time interventions offers a promising direction for scalable bias mitigation in open-world vision–language applications.

# B   Theoretical Supplements

## B.1   Decomposition for CLIP Embeddings

This section provides a compact derivation of the object–context factorization for both visual and textual encoders and explains, from an information-theoretic viewpoint, how interaction terms emerge under the InfoNCE objective which is the training loss for CLIP.

**Notation Restatement.** $X \in \mathcal{X}$ and $Z \in \mathcal{Z}$ denote the object and context random variables respectively. For completeness we write $p(\mathbf{i}, \boldsymbol{x}, \boldsymbol{z})$ for the training distribution and assume the image embedding $f_i(\mathbf{i}) \in \mathbb{R}^d$ and text embedding $f_t(T) \in \mathbb{R}^d$ belong to the Hilbert space $L^2\big(p(\mathbf{i}, \boldsymbol{x}, \boldsymbol{z})\big)$. [7]

Then, we use the shorthand $\mathbb{E}[\cdot \mid X{=}\boldsymbol{x}]$ for $\mathbb{E}_{\mathbf{i} \sim p(\mathbf{i} \mid X=\boldsymbol{x})}[\cdot]$ and similar for $Z$ or $(X, Z)$. All embeddings from dataset are assumed centered: $\mathbb{E}[f_i(\mathbf{i})] = \mathbb{E}[f_t(T)] = \mathbf{0}$.

### B.1.1 Visual Two–Factor Hoeffding Decomposition

Let $g_i(\boldsymbol{x}, \boldsymbol{z}) = \mathbb{E}[f_i(\mathbf{i}) \mid X{=}\boldsymbol{x}, Z{=}\boldsymbol{z}]$ denote the conditional mean embedding obtained from dataset information, which encapsulates the complete statistical memory of the training set for $(\boldsymbol{x}, \boldsymbol{z})$ pairs. Projecting $g$ onto the two one-factor subspaces $\{g_i(\boldsymbol{x}, \cdot)\}$ and $\{g_i(\cdot, \boldsymbol{z})\}$ of the Hilbert space $L^2(p(\boldsymbol{x}, \boldsymbol{z}))$ yields the object and context main effect functions:

$$\boldsymbol{e}_i(\boldsymbol{x}) = \mathbb{E}[f_i(\boldsymbol{i}) \mid X = \boldsymbol{x}], \qquad \boldsymbol{e}_i(\boldsymbol{z}) = \mathbb{E}[f_i(\boldsymbol{i}) \mid Z = \boldsymbol{z}]. \tag{20}$$

Subtracting the two one–way projections from the joint expectations we obtain the interaction residue $r_i(\boldsymbol{x}, \boldsymbol{z}) = g_i(\boldsymbol{x}, \boldsymbol{z}) - e_i(\boldsymbol{x}) - e_i(\boldsymbol{z})$. By construction, for every fixed $\boldsymbol{x}$ we have

$$\begin{aligned} \mathbb{E}\big[r_i(X, Z) \mid X = \boldsymbol{x}\big] &= \mathbb{E}\big[g_i(\boldsymbol{x}, Z) \mid X = \boldsymbol{x}\big] - e_i(\boldsymbol{x}) - \mathbb{E}\big[e_i(Z)\big] \\ &= e_i(\boldsymbol{x}) - e_i(\boldsymbol{x}) - 0 = 0, \end{aligned} \tag{21}$$

and similarly $\mathbb{E}[r_i \mid Z{=}\boldsymbol{z}] = 0$. Orthogonality follows directly from this property: let $\mathcal{V}_X{=}\{h(X) : h \in L^2(p(\boldsymbol{x}))\}$ and $\mathcal{V}_Z{=}\{k(Z) : k \in L^2(p(\boldsymbol{z}))\}$ be the two one-factor subspaces of $L^2(p(\boldsymbol{x}, \boldsymbol{z}))$. For any $h(X) \in \mathcal{V}_X$, according to Law of Total Expectation:

$$\langle r_i, h(X) \rangle = \mathbb{E}\big[r_i \cdot h(X)\big] = \mathbb{E}\big\{h(X)\,\mathbb{E}\big[r_i \mid X\big]\big\} = 0, \tag{22}$$

and the same argument with $k(Z)$ shows $\langle r_i, \mathcal{V}_Z \rangle = 0$. Hence $r_i$ is orthogonal to $\mathcal{V}_X \cup \mathcal{V}_Z$.

The residual term $\eta_i = f_i(\mathbf{i}) - g_i(X, Z)$ satisfies $\mathbb{E}[\eta_i \mid X, Z] = \mathbf{0}$, which for any function $q_1(X) + q_2(Z) + q_3(X, Z)$ belonging to the direct sum $\mathcal{V}_X \oplus \mathcal{V}_Z \oplus \mathcal{V}_{XZ}$ (with $q_3$ centered in both arguments), the inner product with $\eta_i$ vanishes after taking conditional expectations. It essentially captures higher-order variability orthogonal to all lower-order components (i.e., object, background, and interaction terms).

Consequently the four components $\boldsymbol{e}_i(X)$, $\boldsymbol{e}_i(Z)$, $r_i(X, Z)$, $\eta_i$ are pairwise orthogonal, giving rise to the unique expansion reported in Eq. (1) of the main text:

$$f_i(\mathbf{i}) = \boldsymbol{e}_i(\boldsymbol{x}) + \boldsymbol{e}_i(\boldsymbol{z}) + r_i(\boldsymbol{x}, \boldsymbol{z}) + \eta_i. \tag{23}$$

When the data distribution is *ideal distribution* where the pairs of $(\boldsymbol{x}, \boldsymbol{z})$ appears in dataset evenly, the joint mean embedding factorizes into the sum of margins and $r_i$ collapses to $\mathbf{0}$, i.e. $X \perp Z$ which means object and background are then fully "decoupled" in this model. In practice, $r_i$ measures the extent to which this ideal fails because of systematic co-occurrence.

**What are $f_i(\mathbf{i})$ and $g_i(\boldsymbol{x}, \boldsymbol{z})$.** The mapping $f_i : \mathcal{I} \to \mathbb{R}^d$ is a **deterministic** encoder produced by CLIP after contrastive training. For any concrete image $\mathbf{i}$ it outputs an *instance–level* embedding $f_i(\mathbf{i})$. The quantity $g_i(\boldsymbol{x}, \boldsymbol{z}) = \mathbb{E}\big[f_i(\mathbf{i}) \mid X{=}\boldsymbol{x}, Z{=}\boldsymbol{z}\big]$ is the **conditional mean embedding**: it averages the encoder output over all images in the training distribution that share the same object–context label pair $(\boldsymbol{x}, \boldsymbol{z})$. Formally,

$$g_i(\boldsymbol{x}, \boldsymbol{z}) = \int_{\mathcal{I}} f_i(\mathbf{i})\, p(\mathbf{i} \mid X{=}\boldsymbol{x}, Z{=}\boldsymbol{z})\, d\mathbf{i}, \tag{24}$$

which can be estimated in practice by empirical averaging across the mini-batch samples carrying the label $(\boldsymbol{x}, \boldsymbol{z})$. Hence $g$ may be viewed as the "prototype" or population center of all *albatross-in-ocean*, *crane bird-in-forest*, … images that the encoder has seen during training. Because the encoder is fixed, the only randomness in $f_i(\mathbf{i})$ comes from the data distribution; conditioning on $(X, Z)$ therefore yields a well-defined mean element $g_i(\boldsymbol{x}, \boldsymbol{z}) \in L^2\big(p(\mathbf{i})\big)$.

---

[7]The $\ell_2$ normalization in CLIP is applied *after* the embedding is fed into the contrastive loss; the Hoeffding expansion applies to the pre-normalized vectors. For clarity, we omit this distinction in symbols.

### B.1.2 Textual Two–Factor Hoeffding Decomposition

Let $g_t(\boldsymbol{x}, \boldsymbol{z}) = \mathbb{E}[f_t(T) \mid X{=}\boldsymbol{x}, Z{=}\boldsymbol{z}]$ denote the conditional mean embedding of the text encoder over all prompts $T{=}(T_x, T_z)$ generated from label pair $(\boldsymbol{x}, \boldsymbol{z})$. Projecting $g_t$ onto the two one-factor subspaces $\{g_t(\boldsymbol{x}, \cdot)\}$ and $\{g_t(\cdot, \boldsymbol{z})\}$ of the Hilbert space $L^2\big(p(\boldsymbol{x}, \boldsymbol{z})\big)$ yields the object and context main effect functions:

$$\boldsymbol{e}_t(\boldsymbol{x}) = \mathbb{E}[f_t(T) \mid X{=}\boldsymbol{x}], \qquad \boldsymbol{e}_t(\boldsymbol{z}) = \mathbb{E}[f_t(T) \mid Z{=}\boldsymbol{z}]. \tag{25}$$

Subtracting these from the joint expectations gives the text interaction remainder $r_t(\boldsymbol{x}, \boldsymbol{z}) = g_t(\boldsymbol{x}, \boldsymbol{z}) - \boldsymbol{e}_t(\boldsymbol{x}) - \boldsymbol{e}_t(\boldsymbol{z})$, which, by construction, $\mathbb{E}[r_t \mid X{=}\boldsymbol{x}] = \mathbb{E}[r_t \mid Z{=}\boldsymbol{z}] = \boldsymbol{0}$ and is therefore orthogonal to both one-way subspaces.

Finally the residual terms $\eta_t = f_t(T) - g_t(X, Z)$ satisfies $\mathbb{E}[\eta_t \mid X, Z] = \boldsymbol{0}$. Collecting these pieces yields the unique orthogonal expansion:

$$f_t(T_x, T_z) = \boldsymbol{e}_t(\boldsymbol{x}) + \boldsymbol{e}_t(\boldsymbol{z}) + \boldsymbol{r}_t(\boldsymbol{x}, \boldsymbol{z}) + \eta_t, \tag{26}$$

which analogous to Eq. (9) for the visual encoder but on the text side.

### B.2 Interaction Amplification under InfoNCE

We now place the two–factor decompositions Eq. (23) and Eq. (26) into the CLIP training loss and give a self-contained argument that, whenever an object–context pair $(\boldsymbol{x}, \boldsymbol{z})$ appears abnormally frequent than other pairs, gradient optimization enlarges the interaction terms $r_i(\boldsymbol{x}, \boldsymbol{z})$ and $r_t(\boldsymbol{x}, \boldsymbol{z})$.

**Score Decomposition.** Insert the two–factor decomposed embeddings into the normalized inner product $S(\mathbf{i}, T) = \langle f_i, f_t(T)\rangle / \tau$ appearing in CLIP's InfoNCE loss. For a "positive" pair, where $\mathbf{i} \sim p(\mathbf{i} \mid X = \boldsymbol{x}, Z = \boldsymbol{z})$ is an image whose true object label is $\boldsymbol{x}$ and context label is $\boldsymbol{z}$, we obtain:

$$
\begin{aligned}
S_{++} &:= S(\mathbf{i}, T_{x,z}) \\
&= \frac{1}{\tau}\Big[\underbrace{\langle \boldsymbol{e}_i(\boldsymbol{x}), \boldsymbol{e}_t(\boldsymbol{x})\rangle + \langle \boldsymbol{e}_i(\boldsymbol{z}), \boldsymbol{e}_t(\boldsymbol{z})\rangle}_{\text{main effects}} + \underbrace{\langle \boldsymbol{e}_i(\boldsymbol{z}), \boldsymbol{e}_t(\boldsymbol{x})\rangle + \langle \boldsymbol{e}_i(\boldsymbol{x}), \boldsymbol{e}_t(\boldsymbol{z})\rangle}_{\text{cross–alignment}} \\
&\quad + \underbrace{\langle \boldsymbol{e}_i(\boldsymbol{x}) + \boldsymbol{e}_i(\boldsymbol{z}), r_t\rangle + \langle \boldsymbol{e}_t(x) + \boldsymbol{e}_t(\boldsymbol{z}), r_i\rangle + \langle r_i, r_t\rangle}_{\text{interaction}} \\
&\quad + \underbrace{\langle \eta_i, f_t\rangle + \langle \eta_t, f_i\rangle}_{\text{residual}}\Big].
\end{aligned}
\tag{27}
$$

The two one-factor hard negatives $(\boldsymbol{x}, \boldsymbol{z}')$ and $(\boldsymbol{x}', \boldsymbol{z})$ share exactly one main effect vector with the positive sample, hence $S_{+-} = S(\mathbf{i}_{\boldsymbol{x},\boldsymbol{z}'}, T_{\boldsymbol{x},\boldsymbol{z}'})$ and $S_{-+} = S(\mathbf{i}_{\boldsymbol{x}',\boldsymbol{z}}, T_{\boldsymbol{x}',\boldsymbol{z}})$ differ from $S_{++}$ primarily in the interaction terms. Explicitly,

$$
\begin{aligned}
S_{++} - S_{+-} = \frac{1}{\tau}\Big[ &\langle \boldsymbol{e}_i(\boldsymbol{z}), \boldsymbol{e}_t(\boldsymbol{z})\rangle - \langle \boldsymbol{e}_i(\boldsymbol{z}'), \boldsymbol{e}_t(\boldsymbol{z}')\rangle \\
&+ \langle \boldsymbol{e}_i(\boldsymbol{z}) - \boldsymbol{e}_i(\boldsymbol{z}'), \boldsymbol{e}_t(\boldsymbol{x})\rangle + \langle \boldsymbol{e}_t(\boldsymbol{z}) - \boldsymbol{e}_t(\boldsymbol{z}'), \boldsymbol{e}_i(\boldsymbol{x})\rangle \\
&+ \langle \boldsymbol{e}_i(\boldsymbol{z}), r_t(\boldsymbol{x}, \boldsymbol{z}) - r_t(\boldsymbol{x}, \boldsymbol{z}')\rangle + \langle \boldsymbol{e}_t(\boldsymbol{z}), r_i(\boldsymbol{x}, \boldsymbol{z}) - r_i(\boldsymbol{x}, \boldsymbol{z}')\rangle \\
&+ \langle r_i(\boldsymbol{x}, \boldsymbol{z}), r_t(\boldsymbol{x}, \boldsymbol{z}) - r_t(\boldsymbol{x}, \boldsymbol{z}')\rangle + \varepsilon\Big].
\end{aligned}
\tag{28}
$$

where $\varepsilon$ is a higher-order residual term about $\eta_i$ and $\eta_t$. Thus the **margin** separating the positive from its hardest negatives is governed by both the cross–alignment and interaction components. A similar expression holds for $S_{++} - S_{-+}$.

**InfoNCE Gradient on $r_i(\boldsymbol{x}, \boldsymbol{z}), r_t(\boldsymbol{x}, \boldsymbol{z})$.** For a mini-batch data $\mathbb{D}$, minimizing the InfoNCE loss: [8] $-\big[S_{++} - \log\sum_{(\boldsymbol{x}',\boldsymbol{z}')\in\mathbb{D}} \mathrm{e}^{S(\mathbf{i}, T')}\big]$ yields the gradient

$$
\begin{aligned}
\nabla_{r_i}\mathcal{L}_{\text{nce}} &= -\frac{1}{\tau}\big[r_t(\boldsymbol{x}, \boldsymbol{z}) - \alpha_{t,+-}\, r_t(\boldsymbol{x}, \boldsymbol{z}') - \alpha_{t,-+}\, r_t(\boldsymbol{x}', \boldsymbol{z})\big], \\
\nabla_{r_t}\mathcal{L}_{\text{nce}} &= -\frac{1}{\tau}\big[r_i(\boldsymbol{x}, \boldsymbol{z}) - \alpha_{i,+-}\, r_i(\boldsymbol{x}, \boldsymbol{z}') - \alpha_{i,-+}\, r_i(\boldsymbol{x}', \boldsymbol{z})\big].
\end{aligned}
\tag{29}
$$

---

[8]To facilitate display, $T'$ consist of $T_{x'}$ and $T_{z'}$

where $\alpha_{i,\pm\mp}$ and $\alpha_{t,\pm\mp}$ are softmax weights for the two hardest negatives for visual and textual model. If $(\boldsymbol{x}, \boldsymbol{z})$ occurs more frequently than $(\boldsymbol{x}, \boldsymbol{z}')$ or $(\boldsymbol{x}', \boldsymbol{z})$ in data distribution, then $\alpha_{+-}, \alpha_{-+}$ are small and the gradient term $-r_t$ dominates, pushing $r_i$ in the $+r_t$ direction. By symmetry, an analogous expression holds for $\nabla_{r_t} \mathcal{L}_{\text{nce}}$. Iterated SGD steps therefore inflate the terms $r_i(\boldsymbol{x}, \boldsymbol{z})$ and $r_t(\boldsymbol{x}, \boldsymbol{z})$ until the margin $S_{++} - \max\{S_{+-}, S_{-+}\}$ is large enough to satisfy the batch softmax.

## B.3 Cross–modal Attraction on Co-occurring Context and Object under InfoNCE

In addition to the interaction gradients on $r_i, r_t$, InfoNCE also induces cross–alignment gradients on the one–way main effects $\boldsymbol{e}_i(\boldsymbol{z})$ and $\boldsymbol{e}_t(\boldsymbol{x})$ whenever $(\boldsymbol{x}, \boldsymbol{z})$ co-occurs more frequently than its one-factor negatives $(\boldsymbol{x}, \boldsymbol{z}')$ and $(\boldsymbol{x}', \boldsymbol{z})$.

Recall that one of the positive score cross–alignment term $\langle \boldsymbol{e}_i(\boldsymbol{z}), \boldsymbol{e}_t(\boldsymbol{x}) \rangle$ and the negative ones for $(\boldsymbol{x}', \boldsymbol{z})$: $\langle \boldsymbol{e}_i(\boldsymbol{z}), \boldsymbol{e}_t(\boldsymbol{x}') \rangle$. All other negatives either involve $\boldsymbol{z}' \neq \boldsymbol{z}$ (do not dependence on $\boldsymbol{e}_i(\boldsymbol{z})$) or only marginal effects. For a mini-batch data, minimizing the InfoNCE loss and its gradient projected onto $\boldsymbol{e}_i(\boldsymbol{z})$ is

$$
\nabla_{\boldsymbol{e}_i(\boldsymbol{z})} \mathcal{L}_{\text{nce}} = -\frac{1}{\tau} \Big[ \boldsymbol{e}_t(\boldsymbol{x}) + \boldsymbol{e}_t(\boldsymbol{z}) \\
- \alpha_{+-} \big( \boldsymbol{e}_t(\boldsymbol{x}) + \boldsymbol{e}_t(\boldsymbol{z}') \big) - \alpha_{-+} \, \boldsymbol{e}_t(\boldsymbol{z}) + \sum_{T' \notin \{T_{++}, T_{+-}, T_{-+}\}} \alpha_{T'} \, \boldsymbol{e}_t(z_{T'}) \Big]. \tag{30}
$$

Since $\alpha_{i,\pm\mp}$ is the softmax weight on the hardest negative $(\boldsymbol{x}, \boldsymbol{z}')$ and $(\boldsymbol{x}', \boldsymbol{z})$, when $(\boldsymbol{x}, \boldsymbol{z})$ is a high-PMI (high -frequency co -occurrence) pair $\alpha_{i,\pm\mp} \ll 1$. The dominant terms are $-\boldsymbol{e}_t(\boldsymbol{x})$, so the net gradient points approximately toward $+\boldsymbol{e}_t(\boldsymbol{x})$.

Symmetrically, $\nabla_{\boldsymbol{e}_t(\boldsymbol{x})} \mathcal{L}_{\text{nce}}$ contains $-\boldsymbol{e}_i(\boldsymbol{z}) + \beta_{-+} \, \boldsymbol{e}_i(\boldsymbol{z})$, also pulling $\boldsymbol{e}_t(\boldsymbol{x})$ toward $+\boldsymbol{e}_i(\boldsymbol{z})$ in case of high-PMI pair. Hence frequent co-occurrences induce a **cross–modal attraction** between background main effect $\boldsymbol{e}_i(\boldsymbol{z})$ and object main effect $\boldsymbol{e}_t(\boldsymbol{x})$, explaining why the background vector can align with the object dictionary entry and thereby raise the spurious score $\langle \boldsymbol{e}_i(\boldsymbol{z}), f_t(T_x) \rangle$ in Eq. (4).

These cross–alignment gradients demonstrate that InfoNCE training actively aligns the background main effect $\boldsymbol{e}_i(\boldsymbol{z})$ with the object prompt embedding $\boldsymbol{e}_t(\boldsymbol{x})$ whenever $(\boldsymbol{x}, \boldsymbol{z})$ co-occurs frequently. Together with the interaction amplification on $r_i, r_t$, this mechanism explains why the spurious shortcut $\mathcal{I} \rightarrow Z \rightarrow Y$ grows stronger under frequent co-occurrence and underpins the observed hallucination bias in novel scenes.

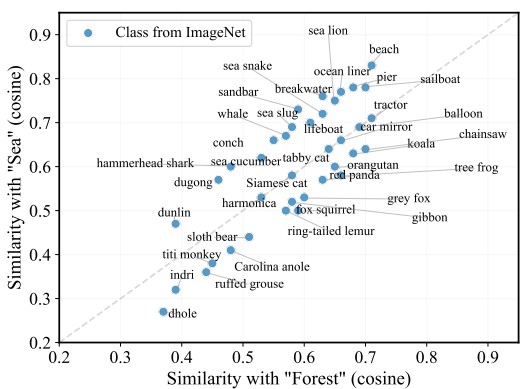

Figure 7: **The illustration of cross–modal attraction.** It compares the textual similarity of two scenes and a part of objects from ImageNet, shown by using $f_t(T_z)$ as a bridge.

## B.4 Information Entropy Perspective for Co-occurrence Shortcuts.

The InfoNCE loss minimizes a tractable lower bound of the mutual information $I(X; Z)$ between objects and scenes:

$$
I(X; Z) \geq \log(K+1) - L_{\text{InfoNCE}}(\theta), \tag{31}
$$

where $K$ denotes the number of negative samples per positive pair (typically $K = N-1$ for a batch size of $N$), it therefore implies that training maximizes the average pointwise mutual information across observed pairs.

$$
I(X; Z) = \mathbb{E}_{p(\boldsymbol{x}, \boldsymbol{z})}[\text{PMI}(\boldsymbol{x}, \boldsymbol{z})] = \mathbb{E}_{p(\boldsymbol{x}, \boldsymbol{z})} \Big[ \log \frac{p(\boldsymbol{x}, \boldsymbol{z})}{p(\boldsymbol{x}) p(\boldsymbol{z})} \Big]. \tag{32}
$$

The CLIP score in Eq. (27) can be decomposed into three main parts: one-way terms, cross-modal terms, and interaction term involving residuals. These components are directly affected by the local $\text{PMI}(\boldsymbol{x}, \boldsymbol{z})$.

When $(\boldsymbol{x}, \boldsymbol{z})$ is a frequently co-occurring object–scene pair (i.e., high $\text{PMI}(x, z)$), the InfoNCE gradient includes attractive forces:

$$-\nabla_{r_i} L \propto r_t, \quad -\nabla_{r_t} L \propto r_i \quad \Rightarrow \quad r_i(\boldsymbol{x}, \boldsymbol{z}), \ r_t(\boldsymbol{x}, \boldsymbol{z}) \uparrow$$

Meanwhile, the cross-modal alignment terms are also reinforced:

$$-\nabla_{\boldsymbol{e}_i(\boldsymbol{z})} L \propto \boldsymbol{e}_t(\boldsymbol{x}), \quad -\nabla_{\boldsymbol{e}_t(x)} L \propto \boldsymbol{e}_i(\boldsymbol{z}) \quad \Rightarrow \quad \langle \boldsymbol{e}_i(\boldsymbol{z}), \boldsymbol{e}_t(\boldsymbol{x}) \rangle \uparrow$$

This "double boost" both magnifies the margin between positives and hard negatives, and strengthens the shortcut path which is a key mechanism underlying hallucination under co-occurrence bias.

In contrast, when $(\boldsymbol{x}, \boldsymbol{z})$ is rare or anti-correlated, the PMI becomes negative or zero, causing the attractive gradient to vanish or reverse. The optimizer keeps:

$$r_i(\boldsymbol{x}, \boldsymbol{z}), r_t(\boldsymbol{x}, \boldsymbol{z}) \to 0, \quad \boldsymbol{e}_i(\boldsymbol{z}) \perp \boldsymbol{e}_t(\boldsymbol{x}),$$

On the other hand, if pairs come with an almost fair and uniform data distribution, it also allows the model to learns a decoupled, object-centric representation.

In conclusion, by maximizing the expectation of PMI, InfoNCE selectively increases interaction energy and cross-modal alignment only for high-PMI object–scene pairs. These components including $r_i$, $r_t$, and $\langle \boldsymbol{e}_i(\boldsymbol{z}), \boldsymbol{e}_t(\boldsymbol{x}) \rangle$ serve as faithful proxies for dataset co-occurrence bias. When the joint distribution $p(\boldsymbol{x}, \boldsymbol{z})$ is decomposed by independence, the learned representation is robust and object-aligned; but when $p(\boldsymbol{x}, \boldsymbol{z})$ is skewed, InfoNCE implicitly injects shortcut dependencies exactly where the bias is strongest.

## B.5  Prompt-Induced Mitigation of Bias

This section shows formally how appending an explicit scene token $T_z$ blocks the spurious causal path $\mathcal{I} \to Z \to Y$, shown in Fig. 3, and also characterizes the effect of an imprecise or incorrect $T_z'$.

**Causal-Intervention View for Prompt-Based Method.** Let $Z$ be the latent scene label inferred from image $\mathcal{I}$, and $Y$ the target class. In the object-only regime the model effectively computes the probability of class $c$:

$$p(Y \mid \mathcal{I}, T_{c_x}) \;=\; \sum_{z \in \mathcal{Z}} p(Y \mid \mathcal{I}, T, Z = \boldsymbol{z}) \, p(Z = \boldsymbol{z} \mid \mathcal{I}), \tag{33}$$

so the pathway $\mathcal{I} \to Z \to Y$ remains open and can introduce spurious hallucination whenever $p(Z \mid \mathcal{I})$ favors classes co-occurring with certain scenes.

By appending $T_z$, we *intervene* on $Z$, clamping it to the observed value:

$$p(Y \mid \mathcal{I}, T_c, T_z) \;=\; p\big(Y \mid \mathcal{I}, T_c, \, Z = \boldsymbol{z}\big) \times p(Z = \boldsymbol{z} \mid \mathcal{I}, T_c, T_z),$$

where $p(Z = \boldsymbol{z} \mid \mathcal{I}, T_c, T_z) = 1$. This removes the marginalization over $Z$ and thus *blocks* the $\mathcal{I} \to Z \to Y$ shortcut entirely, leaving only the legitimate $\mathcal{I} \to X \to Y$ path.

However, if instead one uses an imprecise or incorrect scene token $T_{z'} \neq T_z$, the intervention locks $Z$ at the wrong value. The causal path is still blocked, but now the model conditions on $Z = T_{z'}$ which mismatches the true scene, shifting probability mass toward classes co-occurring with $T_{z'}$ and potentially introducing new misclassifications.

As the experiment in Fig. 8 demonstrates, using a simple object-only $T_0$ fails on both object predictions. The model recovers the desired predictions when using the full scene descriptions $T_2^{bird}$(*in a forest with dense vegetation and massy stones on riverbank*) and $T_2^{cat}$(*in the ocean with gentle waves and surrounded by green hills and clear sky*), whereas the situation only improves to some extent when using the vague descriptions $T_1^{cat}$(*with trees*) and $T_1^{cat}$(*in water*). Alarmingly, the use of wrong descriptions can make the model predictions worse and introduce new misclassification targets, i.e., $T_3^{bird}$(*in the zoo with lush green grass*), $T_4^{bird}$(*in a foggy swamp, wading through murky water*) and $T_3^{cat}$(*in the art gallery with a starry oil painting on the wall*), $T_4^{cat}$(*in the zoo with lush green grass*).

## B.6  Proof of the Direct-Effect Decomposition

We provide the algebraic details omitted from the main text, showing how the residual stream unrolls into additive direct-effect terms and why all LayerNorm (LN) operations can be absorbed into a single projection matrix $R$ up to a constant bias.

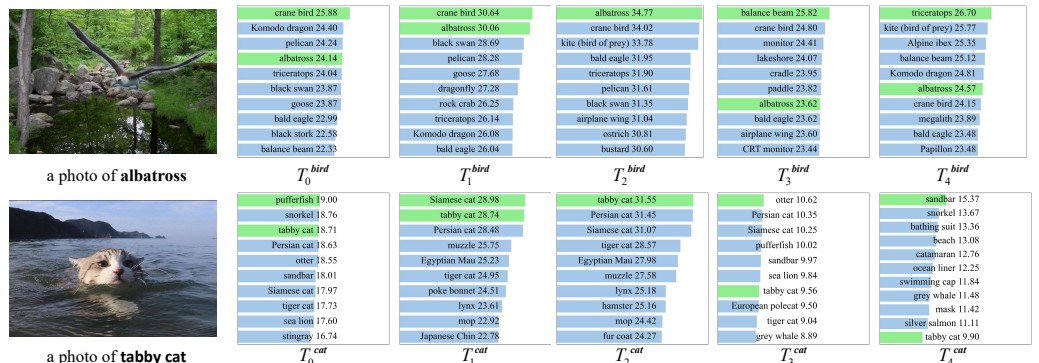

Figure 8: **Impact of context-match level in $T_z$ on prediction scores.** Analysis performed on CLIP (ViT-B-16) for two images about albatross and tabby cat under novel contextual conditions. Five kinds of prompt variants were tested: $T_0$ is baseline (object-only, no context); $T_1$ and $T_2$ are ambiguous and precise scene descriptions; $T_3$ and $T_4$ are incorrect or even adversarial scene descriptions

**Residual Stream Induction** Let $t^0_{cls}, t^0_1, \ldots, t^0_N$ be the token matrix $\mathbf{t}^0 \in \mathbb{R}^{(N+1) \times d}$ fed to the first Transformer block. Each layer $l \in \{1, \ldots, L\}$ consists of:

$$\hat{\mathbf{t}}^l = \mathrm{MSA}^l(\mathrm{LN}^l_1(\mathbf{t}^{l-1})) + \mathbf{t}^{l-1}, \quad \mathbf{t}^l = \mathrm{MLP}^l(\mathrm{LN}^l_2(\hat{\mathbf{t}}^l)) + \hat{\mathbf{t}}^l. \tag{34}$$

Let the residual contributions applying the shared projection $P$, and substitute the first into the second, then the Eq. 34 become:

$$P\,[\mathbf{t}^l]_{\mathrm{cls}} = P\,[\mathbf{t}^{l-1}]_{cls} + P[\mathrm{MSA}^l(\mathrm{LN}^l_1(\mathbf{t}^{l-1}))]_{\mathrm{cls}} + P[\mathrm{MLP}^l(\mathrm{LN}^l_2(\hat{\mathbf{t}}^l))]_{\mathrm{cls}}. \tag{35}$$

By simple induction), we obtain:

$$P\,[\mathbf{t}^L]_{\mathrm{cls}} = P\,t^0_{\mathrm{cls}} + P\sum_{l=1}^{L}[\mathrm{MSA}^l(\mathrm{LN}^l_1(\mathbf{t}^{l-1}))]_{\mathrm{cls}} + P\sum_{l=1}^{L}[\mathrm{MLP}^l(\mathrm{LN}^l_2(\hat{\mathbf{t}}^l))]_{\mathrm{cls}}. \tag{36}$$

Noting that $P\,[\mathbf{t}^L]_{cls} = f_i(\mathbf{i})$ recovers Eq. (7) and thus exhibits the desired additive decomposition into CLS, MSA, and MLP direct effects.

**Absorbing the Pre-projection LayerNorm** Most CLIP variants apply an LN after $\mathbf{t}^L$ but before the projection layer,this term gives:

$$\mathrm{LN}(t) = \underbrace{\frac{\gamma}{\sqrt{\sigma^2 + \epsilon}}}_{\mathrm{I}} \cdot t - \underbrace{\left(\frac{\mu\gamma}{\sqrt{\sigma^2 + \epsilon}} - \beta\right)}_{\mathrm{II}}. \tag{37}$$

where $t \in \mathbb{R}^N$ is token, $\mu_l, \sigma_l$ are the mean and standard deviation, and $\gamma, \beta$ are learned vectors. We absorb the scale I into the projection P and distribute the constant bias II equally to every direct-effect term in Eq. (36). Since II is image-independent, this constant shift changes neither cosine similarities nor downstream zero-shot decisions.

**Justification of the MSA Dominated Semantic Effect** We follow mean-ablation way: each component (CLS, all MLPs, or a set of MSAs) is replaced by its batch mean, leaving the remainder of the network untouched. Formally, for a batch $\mathbb{D}$ of size D and we substitute a component vector:

$$\boldsymbol{x} \leftarrow \frac{1}{D}\sum_{\boldsymbol{x}' \in \mathbb{D}} \boldsymbol{x}'. \tag{38}$$

From Tab. 5, while the decomposition in Eq. (36) is algebraically exact, not all components contribute equally to semantic alignment with the textual prompt in zero-shot classification. The initial class token $t^0_{\mathrm{cls}}$ acts as a global visual prior and does not encode class-specific cues. Empirically, its mean-ablation results weakly decreases (around $0.5\%$ on ImageNet)), suggesting that it plays a stabilizing but non-discriminative role. The MLP blocks, which apply non-linear transformations to

Table 5: **Mean-ablation results on CLIP–ViT components.** We evaluate zero-shot top-1 accuracy on ImageNet by ablating different architectural components in four ViT structures. Each ablation [34] replaces the corresponding residual stream component (e.g. class token, MLPs, MSAs) with its mean over the batch, in Eq. (38).

| Model | Base accuracy | Class token ablation | MLPs ablation | Class token + MLPs | MSA ablation |
|-------|---------------|----------------------|---------------|--------------------|--------------| 
| ViT-B/32 | 65.63 | 64.96 | 64.52 | 64.27 | 0.18 |
| ViT-B/16 | 68.58 | 68.16 | 67.73 | 67.09 | 0.13 |
| ViT-L/14 | 74.01 | 73.72 | 73.22 | 73.15 | 0.05 |
| ViT-H/14 | 76.40 | 75.88 | 75.36 | 75.28 | 0.03 |

each token independently, serve primarily to normalize and rescale features. They do not aggregate new semantic content and result weakly decreases (around $1.0\%$ on ImageNet)).

In contrast, the MSA blocks perform dynamic, content-dependent token mixing. For each layer and attention head, patch tokens that are semantically aligned with the class description receive higher attention scores and are explicitly pooled into the CLS token. This operation introduces the main class-relevant signal into the residual stream. Quantitatively, when ablated for MSAs, it leads to the most significant drop in zero-shot classification accuracy. This clearly indicates that the MSA component is the principal driver of semantic alignment in CLIP–ViT.

Thus, in the way of Eq. (9), we isolate the semantically meaningful path through which each patch contributes to the final image embedding, and excludes the residuals (class token and MLPs) whose contributions are either generic or semantically neutral.

## B.7 The Proof of Conditional Expectation as the Minimum-MSE Predictor

Without loss of generality, this section focuses on $\mathcal{C}(z)$ as our object of study. The proof leads to the following conclusions. The vector $\mathcal{C}(z)$ is the unique minimizer of mean-squared error, the centroid of the conditional background manifold, making Eq. (11) the optimal first-order solution in both theoretical senses.

**Notation.** Let $(\Omega, \mathcal{F}, \mathbb{P})$ be a probability space and

$$X : \Omega \to \mathbb{R}^D, \qquad X \in L^2(\Omega) \ \big(\text{i.e. } \mathbb{E}\|X\|_2^2 < \infty\big).$$

Let $\mathcal{G} \subset \mathcal{F}$ be a sub-$\sigma$–algebra that represents the information available to a predictor. A map $a : \Omega \to \mathbb{R}^D$ is $\mathcal{G}$-*measurable* when it depends only on $\mathcal{G}$.

**Orthogonality (Pythagoras) Identity.** For every $\mathcal{G}$-measurable $a$,

$$\mathbb{E}\big\|X - a\big\|_2^2 \ = \ \mathbb{E}\big\|X - \mathbb{E}[X \mid \mathcal{G}]\big\|_2^2 + \big\|a - \mathbb{E}[X \mid \mathcal{G}]\big\|_{L^2}^2. \tag{39}$$

The first term is the irreducible error; the second is the non-negative excess error that vanishes only if $a = \mathbb{E}[X \mid \mathcal{G}]$. Hence

$$a^\star := \mathbb{E}[X \mid \mathcal{G}]$$

is the **unique** minimiser of MSE among all $\mathcal{G}$-measurable predictors [43].

**Instantiation for Counterfactual Embeddings.** Let $v(\mathbf{i}) \in \mathbb{R}^D$ be the image embedding of input $\mathbf{i}$. Define:

$$X = f_i(\mathbf{i}), \qquad \mathcal{G} = \sigma(G = z) \ (\text{token is background}).$$

Applying (39) yields the centroid of the conditional background manifold

$$\mathcal{C}(z) := \mathbb{E}\big[\, f_i(\mathbf{i}) \mid G = z\big] = \int_{\mathbb{R}^D} \tau \ p_{[f_i(\mathbf{i})|G=z]}(\tau \mid z) \, d\tau. \tag{40}$$

which uniquely minimizes:

$$\mathbb{E}\big\|f_i(\mathbf{i}) - a\big\|_2^2, \qquad a \in L^2(\Omega), \ a \text{ is } \mathcal{G}\text{-measurable}.$$

Equivalently, after sampling for Eq. (40), each token can be written as $\mathbb{E}[v_j(\mathbf{i}) \mid G = z]$. And, the summation is composed of $\mathcal{C}(z)$. Thus, when only the background information $G = z$ is known, $\mathcal{C}(z)$ is the **MMSE-optimal** first-order mean approximation to the true embedding $v(\mathbf{i})$. This provides a principled baseline for constructing counterfactual representations.

# C Details of the Experiment Setting

## C.1 Dataset Description

We evaluate our methods on four widely adopted benchmark datasets specifically designed to assess contextual robustness under spurious correlations and distribution shifts. These datasets span synthetic and real-world biases and provide controlled setups for evaluating zero-shot generalization under strong object–context entanglement:

**Waterbirds** [8]. It constructed by compositing bird foregrounds from CUB-200-2011 with natural scenes from Places [36], focuses on background–label correlation. It comprises a total of 10,589 images, including 4,795 training images and 5,794 test images, categorized into the classes: waterbirds and landbirds." In the training set reflecting reality, over $95\%$ of waterbirds appear in water backgrounds and landbirds in land backgrounds, presenting strong spurious context–class associations. The test set balances background distributions to evaluate generalization beyond shortcuts.

**UrbanCars** [9]. It derived from Stanford Cars, Places, and LVIS datasets, introduces multi-bias settings by synthetically composing each image with a car object, a background (urban or rural), and a co-occurring object (e.g., fire hydrants, cows). The training set contains 8,000 images, with strong three-way correlations between class, background, and co-occurring object. Both validation and test sets contain 1000 balanced samples where the spurious cues have extra tags which enables fine-grained analysis of shortcut reliance and interaction among multiple biases.

**COCO-GB** [10]. It reorganized from MS-COCO [44], targets gender–context co-occurrence biases in image captioning. Based on gender annotation inferred from captions and manual verification, two versions are constructed: v1 includes a test set of images with equal male and female representation across balanced contexts for fairness analysis. These setups expose hallucinated gender attributions when models rely excessively on contextual priors rather than visual cues, shown in Fig. 2.

**NICO** [11]. It is a real-world dataset contains over 25,000 images spanning 19 object classes (10 animals, 9 vehicles) across 188 distinct contexts (e.g., "dog on grass", "car in snow"). Each class appears in 10 different environmental or situational contexts. It serves as a benchmark built specifically to study context bias and causal learning, aiming at exploring causal relationships between objects and their contexts (e.g., background, scene, co-occurring objects) in object recognition tasks.

Together, these datasets enable rigorous and controlled evaluation of models' ability to disentangle semantic representations from contextually entangled signals, particularly under zero-shot and distribution-shifted scenarios. Fig. 9 displays a subset of images from those dataset.

## C.2 Implementation Details

We use four CLIP vision backbones: ViT-B/32, ViT-B/16, ViT-L/14, and ViT-H/14, all publicly available via OpenCLIP and pretrained on the LAION-2B-en dataset [26], which contains over 2.3 billion image–text pairs. The pretrained model are: laion2b_s34b_b79k for ViT-B/32, laion2b_s34b_b88k for ViT-B/16, laion2b_s32b_b82k for both ViT-L/14 and laion2b_s32b_b79k for ViT-H/14. Following CLIP's original setting [1], we adopt 80 hand-crafted text prompts per class to support standard zero-shot evaluation.

Then, we implement three versions of CounterfactualCLIP in this paper, distinguished by the source of counterfactual scene embeddings. Each follows a unified architecture and inference procedure, differing only in how background contexts are collected:

**External Scene Variant.** This variant uses scene images sampled from the Places dataset. For each target dataset (e.g., Waterbirds), we manually select 16 relevant scene categories based on background diversity observed in the data (see Tab. 6). For each category, 50 images are randomly drawn from Places, yielding a fixed pool of 800 candidate backgrounds. At inference time, for each test image, We employed two constraints to uniformly select a total of $M$ scenes from 16 relevant scene categories in the scene pool: (1) maximized scene dissimilarity with the original input; and (2) Maximized visual dissimilarity with the foreground object class. The selected scenes are embedded via the CLIP image encoder and used to synthesize counterfactual image representations.

**Internal Scene Variant.** Here, the candidate pool is constructed online using other samples in the same test-time batch. We exclude the current image and select $M$ candidates from remaining batch

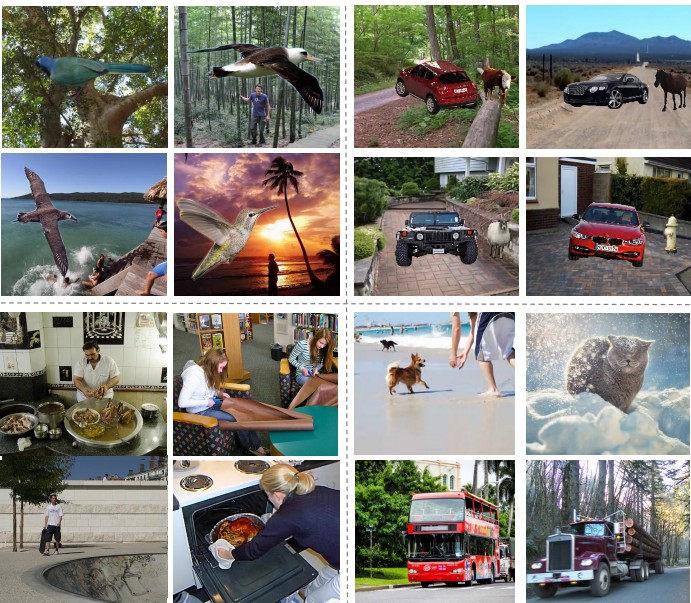

Figure 9: **Examples from four datasets: Waterbirds, UrbanCars, COCO-GB and NICO.**

Table 6: **Selected scene categories from the Places dataset for each target dataset in the External Scene Variant.**

| Dataset | Scene Categories (16 total) |
| --- | --- |
| **Waterbirds** | marsh, pond, lake-natural, river, swamp, creek, canal-natural, fishpond, forest-broadleaf, bamboo_forest, orchard, field-cultivated, field-wild, pasture, farm, forest_path |
| **UrbanCars** | street, downtown, parking_lot, parking_garage-outdoor, highway, gas_station, viaduct, crosswalk, field-cultivated, field_road, forest_road, village, farm, pasture, barn |
| **COCO-GB** | living_room, bedroom, kitchen, bathroom, dining_room, office, classroom, hotel_room, restaurant, shopping_mall-indoor, bus_station-indoor, supermarket, park, street, playground |
| **NICO** | beach, bridge, desert_road, forest_path, river, snowfield, street, residential_neighborhood, home_office, mountain_snowy, pasture, airplane_cabin, train_station-platform, garage-outdoor, amphitheater, corral |

samples by Eq. (11), applying the same semantic dissimilarity filters as the external variant. This variant is more dynamic and requires no external data but depends on batch diversity.

For both external and internal variants $M$ contexts are selected uniformly across categories (external) or available pool entries (internal) after dissimilarity filtering. This helps maintain scene diversity and avoid sampling dominant or overly similar backgrounds.

**Virtual Scene Variant.** This variant constructs a purely text-based counterfactual pool. For each dataset, we generate 400 diverse scene descriptions using a large language model. The prompt is phrased as:

*Generate [NUMBER] unique, single-sentence scene descriptions that together span the full range of environments found in [DATASET], without referring to any [CLASS] specific content.*

These descriptions, partly shown in Tab. 7, are then encoded via the CLIP text encoder to obtain virtual scene embeddings, which act as background surrogates during counterfactual synthesis. This variant is lightweight, label-agnostic, and easily extensible. In the experiment, we sample $M$ background scenes from a candidate pool of size $B = 400$ for each image to synthesize the virtual counterfactuals,

Table 7: **Examples of generated scene descriptions for the Virtual scene variant on Waterbirds and COCO-GB.**

| | Waterbirds | | COCO-GB |
|---|---|---|---|
| # | Scene Description | # | Scene Description |
| 1 | A clear coast near a tidal pool in the early morning. | 1 | A sidewalk café with metal chairs and small round tables facing the street. |
| 2 | A misty floodplain near a gravel path on a rainy day. | 2 | A hospital room with medical equipment, a monitor, and a bed by the window. |
| 3 | A shimmering marsh near a grassy bank at dawn. | 3 | A forest clearing surrounded by tall trees and a carpet of fallen leaves. |
| 4 | A glassy wetland near a coastal bluff on a rainy day. | 4 | A classroom with desks arranged in rows and a whiteboard covered in diagrams. |
| 5 | A foggy sea near a gravel path on a rainy day. | 5 | A theater stage with drawn curtains and lights focused on the center. |
| 6 | A shaded lagoon near a dune on a rainy day. | 6 | A public plaza with a fountain in the center and benches arranged around it. |
| 7 | A tranquil river near a reed bed under midday sun. | 7 | A mountain hiking trail winding between large boulders and dense pine trees. |
| 8 | A glassy lake near a wooden pier in the early morning. | 8 | A shopping mall interior with decorative plants and store signs on every side. |

for instance, with $M = \{270, 270, 190, 90\}$ for ViT-B/32, ViT-B/16, ViT-L/14, and ViT-H/14 on Waterbirds, respectively.

**Additional Hyper-parameters.** During counterfactual synthesis we blend the object and background embeddings with a fusion weight $\alpha$. We find $\alpha$ selects 0.5-0.7 yields the best trade-off. The coefficient $\lambda$ in Eq. (19) regulates how much of the learned object–context interaction is retained in the final score; values 0.6-0.8 consistently work well. In Eq. (15), we introduce an additional factor $\hat{\lambda}$ to control the malignant hallucination term, normally selecting 1 unless stated otherwise.

When forming a counterfactual image embedding in Eq. (11) we optionally discard token contributions whose probability is below a threshold, which can be tightened for unknown noisy. It sets around 0.3 usually provides a good balance. Finally, to limit inference overhead, the operation of Eq. (18) are applied only to the top-5 classes returned by the initial softmax; all other classes still receive the TDE correction. The intuition stems from the observation that the model only needs to imagine the top-K most confusable categories in alternative contexts to achieve robust predictions. A sensitivity analysis of some of parameters above is presented in Appendix E.3.

# D   Details of the Experiment Results

## D.1   More Results on Waterbirds

Tab. 8 reports per-group accuracy on Waterbirds for all ViT backbones, where each sample is defined by its object label (landbird: LB or waterbird: WB) and background (land: L or water: W). The diagonal cells (LB–L, WB–W) correspond to familiar object–background pairs, whereas off-diagonal cells (LB–W, WB–L) represent the hardest OOD compositions that expose context bias.

Across backbones, vanilla CLIP shows strong degradation on LB–W and WB–L groups. For example, only 34.6% on LB–W with ViT-B/32 confirms its reliance on background cues. CounterfactualCLIP substantially lifts these two worst groups while maintaining or improving the in-distribution pairs. These detailed results above further illustrate that our method suppresses spurious scene shortcuts and restores object-centric decision making.

## D.2   More Results on COCO-GB

Tab. 9 presents the average and subgroup accuracy on COCO-GB v1 across ViT-B/32, ViT-L/14, and ViT-H/14 backbones. We observe that vanilla CLIP suffers from notable gender gaps (e.g.,6.00% on ViT-B/32), indicating that model predictions are influenced by gender–context co-occurrence bias.

Table 8: **Performance on Waterbirds across different ViT backbones.** LB: landbird, WB: waterbird; L: land background, W: water background.

| Backbone | Method | LB-L↑ | LB-W↑ | WB-L↑ | WB-W↑ | Avg.↑ |
|---|---|---|---|---|---|---|
| ViT-B/32 | CLIP-base | 88.47 | 34.55 | 59.03 | 94.39 | 64.88 |
| | Ours(External) | 79.16 | 80.67 | **79.91** | 80.37 | 79.96 |
| | Ours(Internal) | 87.32 | 78.98 | 71.81 | 83.49 | 81.93 |
| | Ours(Virtual) | **90.64** | **82.93** | 65.73 | 79.91 | **83.69** |
| ViT-B/16 | CLIP-base | 95.57 | 63.81 | 50.93 | 85.98 | 77.20 |
| | Ours(External) | 92.90 | 86.39 | 70.87 | 79.13 | 86.40 |
| | Ours(Internal) | 88.16 | 82.88 | **82.71** | **87.54** | 85.43 |
| | Ours(Virtual) | 95.30 | **94.41** | 67.13 | 73.99 | **89.47** |
| ViT-L/14 | CLIP-base | 95.30 | 52.51 | 71.03 | 92.52 | 75.65 |
| | Ours(External) | 92.82 | 93.13 | **86.76** | 82.09 | 91.08 |
| | Ours(Internal) | 93.84 | 89.00 | 85.67 | **90.81** | 90.71 |
| | Ours(Virtual) | **95.61** | 93.97 | 83.02 | 80.84 | **91.94** |
| ViT-H/14 | CLIP-base | 96.27 | 71.65 | 41.28 | 68.85 | 77.55 |
| | Ours(External) | 93.75 | **93.92** | 73.68 | 63.40 | 88.23 |
| | Ours(Internal) | 92.24 | 92.68 | **74.14** | 67.45 | 87.66 |
| | Ours(Virtual) | **97.21** | 91.18 | 68.38 | **70.40** | **88.70** |

Table 9: **Average and gender-specific accuracy (%) across three CLIP backbones on COCO-GB v1.** Gap denotes gender performance differences.

| Backbone | Method | Avg.↑ | Female↑ | Male↑ | Gap↓ |
|---|---|---|---|---|---|
| ViT-B/32 | CLIP | 86.20 | 83.20 | 89.20 | 6.00 |
| | TBD | 86.60 | 82.20 | 91.00 | 8.80 |
| | PC | 86.30 | 84.80 | 87.80 | 3.00 |
| | PC$^+$ | 86.80 | 83.40 | 90.20 | 6.80 |
| | B2T | 86.60 | 81.40 | 91.80 | 10.40 |
| | Ours (External) | **87.30** | **85.80** | 88.80 | **3.00** |
| | Ours (Internal) | 87.00 | 85.40 | 88.60 | 3.20 |
| | Ours (Virtual) | 87.20 | 84.80 | 89.60 | 4.80 |
| ViT-L/14 | CLIP | 91.05 | 90.50 | 91.60 | 1.10 |
| | TBD | 91.60 | 91.00 | 92.20 | 1.20 |
| | PC | 91.80 | 92.60 | 91.00 | 1.60 |
| | PC$^+$ | 91.50 | 90.40 | 92.60 | 2.20 |
| | B2T | 91.50 | 90.00 | 93.00 | 3.00 |
| | Ours (External) | 91.70 | 91.80 | 91.60 | **0.20** |
| | Ours (Internal) | 91.75 | 92.10 | 91.40 | 0.70 |
| | Ours (Virtual) | **92.15** | 92.60 | 91.70 | 0.90 |
| ViT-H/14 | CLIP | 92.70 | 93.40 | 92.00 | 1.40 |
| | TBD | 92.30 | 93.40 | 91.20 | 2.20 |
| | PC | 92.80 | 92.60 | 93.00 | 0.40 |
| | PC$^+$ | 93.10 | 92.60 | 94.00 | 1.40 |
| | B2T | 91.70 | 92.20 | 91.20 | 1.00 |
| | Ours (External) | 92.90 | 92.60 | 93.20 | 0.60 |
| | Ours (Internal) | **93.20** | **93.40** | 93.00 | **0.40** |
| | Ours (Virtual) | 93.00 | 93.20 | 92.80 | **0.40** |

CounterfactualCLIP maintains competitive or better average accuracy while significantly narrowing the gender gap across all backbones. On ViT-L/14, for example, our external variant achieves a gap of only $0.20\%$, and on ViT-H/14, the virtual variant reduces the gap to just $0.40\%$. These results confirm that our method effectively mitigates gender-associated spurious correlations without sacrificing overall performance. Results for ViT-B/16 are reported separately in Tab. 2).

Table 10: **Average and factor-specific accuracy (%) on UrbanCars across three CLIP backbones.** We report accuracy on the original co-occurrence (I.D.), background shifted (BG), and Co-object shifted (Co-Obj) subsets, along with overall average accuracy.

| Backbone | Method | Avg.↑ | I.D.↑ | BG↑ | Co-Obj↑ |
|---|---|---|---|---|---|
| ViT-B/32 | CLIP | 63.73 | 79.60 | 36.40 | 75.20 |
| | TBD | 59.33 | 74.00 | 37.60 | 66.40 |
| | PC | 69.73 | 83.20 | 52.40 | 73.60 |
| | PC$^+$ | 70.93 | 84.40 | 48.00 | 80.40 |
| | B2T | 67.47 | 84.80 | 36.80 | 80.80 |
| | Ours (External) | 71.60 | 84.40 | 50.00 | 80.40 |
| | Ours (Internal) | **72.00** | 86.40 | 48.00 | 81.60 |
| | Ours (Virtual) | 71.07 | **86.80** | 44.00 | **82.40** |
| ViT-L/14 | CLIP | 62.27 | 80.40 | 32.80 | 73.60 |
| | TBD | 55.60 | 66.00 | 44.80 | 56.00 |
| | PC | 62.67 | 77.60 | 45.60 | 64.80 |
| | PC$^+$ | 64.40 | 84.80 | 36.00 | 72.40 |
| | B2T | 61.60 | 79.60 | 32.00 | 73.20 |
| | Ours (External) | 63.07 | 83.20 | 35.60 | 70.40 |
| | Ours (Internal) | 64.40 | 84.00 | 34.80 | **74.40** |
| | Ours (Virtual) | **66.00** | **88.40** | 36.00 | 73.60 |
| ViT-H/14 | CLIP | 62.27 | 84.00 | 34.40 | 68.40 |
| | TBD | 60.53 | 72.80 | 49.60 | 59.20 |
| | PC | 63.20 | 82.00 | 39.60 | 68.00 |
| | PC$^+$ | 64.13 | 82.80 | 38.40 | 71.20 |
| | B2T | 60.93 | 78.40 | 31.20 | 73.20 |
| | Ours (External) | 66.00 | 85.20 | 44.00 | 68.80 |
| | Ours (Internal) | 68.27 | **90.00** | 46.00 | 68.80 |
| | Ours (Virtual) | **68.40** | 88.80 | **47.20** | 69.20 |

## D.3 More Results on UrbanCars

Tab. 10 presents a detailed breakdown on the UrbanCars dataset across ViT-B/32, ViT-L/14, and ViT-H/14 backbones. The three columns (I.D., BG, Co-Obj) respectively represent accuracy on original co-occurring samples, background-shifted samples, and co-object-shifted samples. CLIP shows sharp performance drops under distribution shifts—for example, on ViT-B/32, accuracy drops from $79.6\%$ (I.D.) to only $36.4\%$(BG), highlighting vulnerability to background bias. In contrast, CounterfactualCLIP maintains strong performance across all settings. On ViT-B/32, our internal variant achieves the highest average accuracy ($72.00\%$), while our virtual variant yields the best balance on ViT-H/14, reaching $68.40\%$ average and $47.20\%$ BG accuracy—substantially outperforming CLIP by over 12 points. For ViT-B/16 results, refer to Appendix Tab. 3.

## D.4 More Results on NICO

NICO is a standard benchmark for evaluating contextual robustness, as it includes diverse object categories across varying backgrounds. Tab. 11 and 12 report both average and worst-group accuracy on ViT-B/32 and ViT-B/16. Across both backbones, all three variants of ours methods consistently outperform the baselines in worst-group accuracy, particularly for context-sensitive categories such as "cat" and "sheep." Notably, the virtual variant achieves the best balance between maintaining high average accuracy and significantly improving worst-group performance, highlighting its effectiveness

in mitigating context-induced failures. This improvement stems from ours ability to simulate diverse scene interventions during inference without disrupting semantic alignment.

Table 11: **Average and worst-group accuracy (%) per-category on NICO (ViT-B/32).**

| Metric | Method | airplane | bear | bicycle | bird | boat | bus | car | cat | cow | dog | elephant | helicopter | horse | monkey | motorcycle | rat | sheep | train | truck |
|---|---|---|---|---|---|---|---|---|---|---|---|---|---|---|---|---|---|---|---|---|
| Average | CLIP | 96.19 | 96.44 | 98.76 | 97.28 | 98.96 | 96.60 | 93.45 | 94.04 | 98.17 | 95.17 | 99.75 | 98.23 | 97.88 | 98.57 | 97.07 | 94.06 | 98.16 | 98.54 | 93.45 |
| | TBD | 93.76 | 95.64 | 98.46 | 96.73 | 98.56 | 95.92 | 94.41 | 90.76 | 97.84 | 95.65 | 99.75 | 97.72 | 97.33 | 98.66 | 96.05 | 94.29 | 98.05 | 98.54 | 87.60 |
| | P | 97.78 | 95.83 | 98.52 | 97.47 | 99.14 | 96.40 | 93.74 | 93.12 | 98.25 | 95.59 | 99.83 | 97.42 | 97.80 | 98.66 | 97.13 | 93.82 | 98.16 | 98.41 | 94.05 |
| | PC$^+$ | 97.35 | 95.64 | 98.10 | 97.59 | 99.14 | 96.70 | 93.45 | 93.64 | 98.09 | 94.74 | 99.75 | 97.57 | 97.88 | 98.75 | 97.96 | 94.64 | 98.16 | 98.41 | 93.85 |
| | B2T | 96.72 | 94.54 | 98.46 | 97.22 | 96.44 | 96.79 | 90.75 | 94.69 | 97.92 | 94.80 | 99.66 | 97.87 | 98.35 | 98.84 | 97.13 | 93.94 | 96.85 | 97.49 | 94.15 |
| | Ours(external) | 96.72 | 95.95 | 98.87 | 96.85 | 98.87 | **96.79** | 92.39 | **94.69** | **98.67** | 95.11 | 99.66 | **98.23** | 97.72 | **98.66** | **97.64** | 94.06 | 98.05 | **98.54** | 94.74 |
| | Ours(internal) | 96.72 | 95.15 | 98.70 | 96.60 | 98.60 | 96.70 | 91.81 | 94.63 | **98.67** | 94.98 | 99.66 | 98.16 | 97.80 | 98.57 | 97.96 | **94.76** | 97.94 | 98.41 | **95.04** |
| | Ours(virtual) | **98.10** | **96.62** | **99.11** | 97.28 | 98.83 | 96.70 | 91.62 | 94.63 | 98.59 | 94.68 | 99.66 | 97.79 | 97.80 | 98.48 | 97.13 | 93.24 | 97.94 | 98.28 | 94.94 |
| Worst | CLIP | 87.50 | 88.73 | 96.55 | 92.24 | 97.06 | 66.67 | 77.78 | 73.20 | 94.78 | 93.36 | 97.78 | 96.30 | 92.21 | 93.97 | 89.11 | 84.71 | 93.94 | 95.05 | 70.45 |
| | TBD | 78.57 | 90.08 | 96.03 | 92.24 | 95.59 | 62.22 | 75.00 | 68.18 | 91.07 | 90.71 | 97.78 | 92.59 | 89.61 | 92.24 | 90.10 | 85.88 | 95.45 | 93.48 | 70.45 |
| | P | 91.47 | 88.73 | 95.86 | 92.65 | 97.96 | 64.44 | 80.56 | 72.55 | 94.85 | 94.06 | 97.78 | 94.44 | 91.49 | 94.83 | 89.11 | 83.53 | 93.94 | 95.24 | 79.55 |
| | PC$^+$ | 88.37 | 88.73 | 93.79 | 93.47 | 97.96 | 66.67 | 77.78 | 71.24 | 92.86 | 92.66 | 97.78 | 94.44 | 92.20 | 95.69 | 95.05 | 84.71 | 93.94 | 95.24 | 81.82 |
| | B2T | 89.29 | 84.51 | 94.67 | 90.61 | 89.22 | 71.11 | 72.84 | 77.12 | 94.85 | 92.62 | 97.78 | 95.37 | 92.91 | 96.00 | 90.10 | 86.00 | 92.00 | 95.24 | 85.57 |
| | Ours(external) | 89.29 | **90.14** | 95.86 | 91.84 | **97.96** | 71.11 | 77.78 | 74.51 | **96.27** | 91.88 | 97.78 | **96.93** | 89.36 | **93.97** | 91.09 | 84.00 | 93.94 | **95.24** | 75.00 |
| | Ours(internal) | 89.29 | 88.73 | 95.86 | 91.43 | 97.79 | **71.11** | 77.78 | 74.51 | **96.27** | 91.51 | 97.78 | 96.62 | 89.36 | **93.97** | 92.08 | 85.88 | 93.94 | **95.24** | 77.27 |
| | Ours(virtual) | 91.07 | 91.90 | **96.55** | 92.65 | **97.96** | **71.11** | 77.78 | 74.51 | **96.27** | 90.71 | **97.78** | 96.30 | 89.36 | **93.97** | 91.09 | 82.35 | 93.94 | **95.24** | 77.27 |

Table 12: **Average and worst-group accuracy (%) per-category on NICO (ViT-B/16).**

| Metric | Method | airplane | bear | bicycle | bird | boat | bus | car | cat | cow | dog | elephant | helicopter | horse | monkey | motorcycle | rat | sheep | train | truck |
|---|---|---|---|---|---|---|---|---|---|---|---|---|---|---|---|---|---|---|---|---|
| Average | CLIP | 96.83 | 97.11 | 99.47 | 98.09 | 99.14 | 96.89 | 93.45 | 94.82 | 98.17 | 96.62 | 99.83 | 98.38 | 99.21 | 99.11 | 97.71 | 95.57 | 98.92 | 99.60 | 95.63 |
| | TBD | 89.10 | 95.70 | 98.93 | 97.84 | 94.28 | 96.89 | 94.32 | 94.36 | 98.42 | 96.31 | 99.92 | 97.57 | 98.66 | 98.66 | 94.59 | 95.10 | 98.48 | 98.68 | 92.06 |
| | P | 97.78 | 96.44 | 99.17 | 98.02 | 99.41 | 96.79 | 94.80 | 94.56 | 98.17 | 96.07 | 99.92 | 98.09 | 99.29 | 99.11 | 98.22 | 94.76 | 98.70 | 99.07 | 94.74 |
| | PC($^+$) | 97.10 | 96.50 | 99.17 | 98.33 | 99.28 | 96.79 | 94.22 | 94.89 | 98.17 | 96.13 | 99.92 | 97.94 | 99.21 | 99.11 | 98.20 | 95.45 | 99.02 | 99.07 | 94.84 |
| | B2T | 94.18 | 95.21 | 99.05 | 98.21 | 97.16 | 96.79 | 91.52 | 95.41 | 98.25 | 96.31 | 99.92 | 98.38 | 99.37 | 98.84 | 97.77 | 95.10 | 98.37 | 93.39 | 95.54 |
| | Ours(external) | 97.14 | 97.11 | 99.47 | 98.21 | 99.32 | **98.64** | 87.86 | **97.71** | **98.84** | **98.13** | **100.0** | **99.48** | 99.37 | 99.82 | 97.32 | **97.32** | **99.35** | 99.47 | 95.44 |
| | Ours(internal) | 97.25 | 98.04 | 99.59 | 98.33 | 99.41 | 98.83 | 90.66 | 97.38 | 98.67 | 97.82 | **100.0** | 99.34 | 99.21 | **99.91** | **98.22** | 96.74 | **99.60** | **99.60** | **95.93** |
| | Ours(virtual) | 97.78 | **98.40** | **99.70** | **98.64** | 99.41 | **98.64** | 90.46 | 97.38 | 98.67 | 97.70 | **100.0** | 99.04 | 99.21 | 99.73 | 98.03 | 96.62 | 99.24 | **99.60** | 95.83 |
| Worst | CLIP | 87.50 | 91.90 | 98.22 | 93.88 | 96.94 | 73.33 | 69.44 | 74.51 | 95.52 | 94.25 | 98.55 | 96.30 | 95.74 | 96.55 | 93.07 | 86.05 | 95.45 | 98.41 | 72.73 |
| | TBD | 59.69 | 86.64 | 97.62 | 93.06 | 81.12 | 77.78 | 72.22 | 75.16 | 92.86 | 94.25 | 98.55 | 95.37 | 93.62 | 95.69 | 83.17 | 86.05 | 95.45 | 95.83 | 79.55 |
| | P | 90.70 | 88.66 | 96.45 | 94.29 | 97.96 | 77.78 | 72.22 | 72.55 | 96.27 | 91.95 | 98.55 | 96.30 | 96.45 | 96.55 | 95.60 | 84.88 | 95.45 | 97.83 | 77.27 |
| | PC($^+$) | 91.47 | 89.47 | 95.86 | 94.69 | 96.94 | 77.78 | 77.78 | 74.51 | 94.64 | 94.25 | 98.55 | 96.30 | 96.45 | 96.55 | 95.05 | 84.88 | 95.45 | 96.83 | 79.55 |
| | B2T | 85.71 | 85.02 | 96.45 | 94.69 | 92.16 | 75.56 | 72.84 | 78.43 | 96.27 | 94.44 | 98.55 | 96.55 | 97.78 | 95.69 | 95.05 | 86.05 | 95.45 | 54.17 | 84.54 |
| | Ours(external) | 87.50 | 91.90 | 98.65 | 95.10 | **97.45** | **91.11** | 69.44 | **84.97** | **97.44** | 95.20 | **100.0** | **98.28** | 96.45 | **99.41** | 93.07 | **90.70** | **96.00** | 98.41 | 81.82 |
| | Ours(internal) | 87.50 | 95.14 | 98.22 | 95.10 | **97.45** | **91.11** | 75.00 | 82.35 | 97.01 | 94.46 | **100.0** | 96.55 | 96.45 | **99.41** | 94.06 | 88.37 | **96.00** | 98.41 | 79.55 |
| | Ours(virtual) | 91.07 | **95.55** | **98.82** | 95.92 | **97.45** | **91.11** | 72.22 | 82.35 | 97.01 | 94.46 | **100.0** | 96.55 | 96.45 | 98.70 | 94.06 | 88.37 | **96.00** | 98.41 | 79.55 |

## D.5 Visualization on NICO

We include extended visualization experiment in Fig. 10.

# E More Discussion at CounterfactualCLIP

## E.1 Comparison with Image augmentation Variants under Zero-shot Classification

We further benchmark our method against four classical image augmentation techniques including Mixup, CutMix, AugMix, and Cutout and adapted into counterfactual variants compatible with zero-shot settings. Specifically, instead of retraining, we apply each augmentation multiple times to a test image during inference, and aggregate their prediction logits, obtaining the revisions to prediction. Low-quality augmented samples are filtered [28] to ensure the relevance of counterfactual variations.

Among these, Mixup performs linear interpolation image pairs; CutMix pastes patches from another image; AugMix applies diverse augmentation chains followed by probabilistic blending; and Cutout randomly masks out rectangular regions. All operate at the pixel level, without using pretrained semantic priors. Results in Tab. 13, 14 and 15 show some augmentation variants offer partial

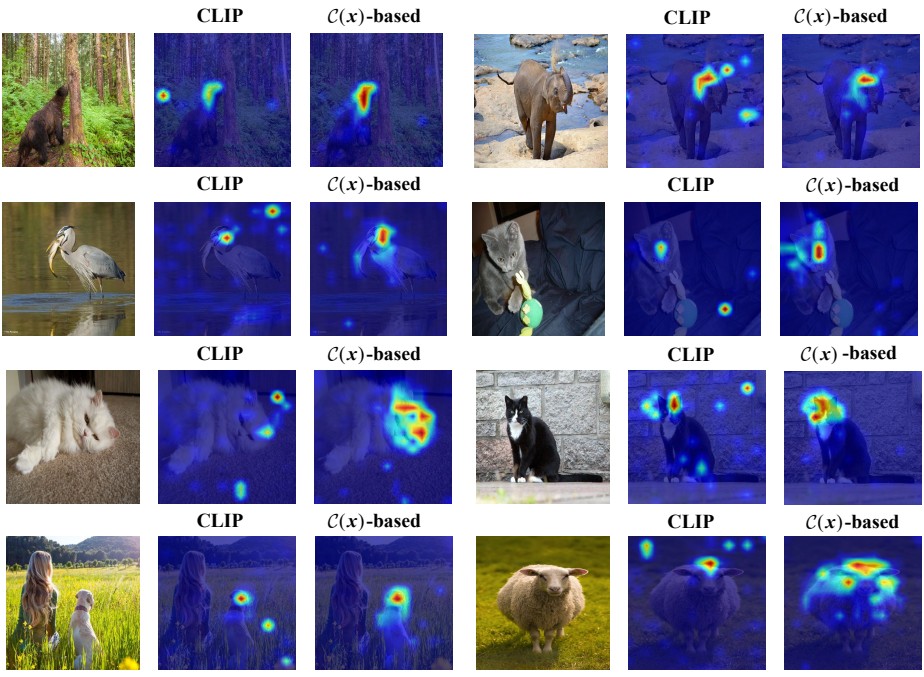

Figure 10: **More Attention Map Revealed by $\mathcal{C}(x)$-based Embeddings on NICO.**

improvements. For example, AugMix reduces the gender gap to $1.6\%$ on COCO-GB and pushes Co-Object accuracy to $78.0\%$ on UrbanCars. However, these improvements are inconsistent and highly dataset-dependent. In contrast, our method consistently outperforms all baselines: the virtual variant improves Waterbirds average accuracy by $+14.1\%$, and narrows COCO gender gap to just $0.8\%$.

Table 13: **Performance comparison with image enhancement variants on Waterbirds (ViT-B/16).**
LB: landbird, WB: waterbird; L: land background, W: water background.

| Method | LB-L↑ | LB-W↑ | WB-L↑ | WB-W↑ | Avg.↑ |
|---|---|---|---|---|---|
| CLIP | 95.57 | 63.81 | 59.03 | 85.98 | 77.20 |
| Mixup | 88.60 | 57.12 | 64.33 | 90.50 | 73.87 |
| CutMix | 95.21 | 57.87 | 47.04 | 85.98 | 74.32 |
| AugMix | 91.53 | 52.28 | 59.50 | 90.03 | 72.54 |
| Cutout | 97.12 | 55.65 | 38.16 | 86.45 | 73.27 |
| Ours (External) | 92.90 | 86.39 | 70.87 | 79.13 | 86.40 |
| Ours (Internal) | 88.16 | 82.88 | 82.71 | 87.54 | 85.43 |
| Ours (Virtual) | 95.30 | 94.41 | 67.13 | 73.99 | 89.47 |

### E.2 Analysis of Module efficiency

As shown in Table 16, the computational cost introduced by our method is minimal, especially compared to diffusion based methods. While our method introduces additional inference steps, we emphasize that all operations are performed in the embedding space, with no image re-encoding or pixel-level augmentation required. CLIP performs a single forward pass through a frozen encoder, and our method preserves this efficiency: it constructs counterfactual embeddings using closed-form computations in Eq. (10), (18) over precomputed token features, followed by lightweight TDE fusion in Eq. (19). All operations involve only matrix multiplications and element-wise computations, without any backpropagation or additional network training. Moreover, since the context candidate pool can be prepared offline, our method remains highly efficient in large-scale data deployment.

Table 14: **Performance comparison with image enhancement variants on COCO-GB v1 (ViT-B/16).** Gap denotes gender performance differences.

| Method | Avg.↑ | Female↑ | Male↑ | Gap↓ |
|---|---|---|---|---|
| CLIP | 88.70 | 85.60 | 91.80 | 6.20 |
| Mixup | 87.30 | 91.00 | 83.60 | 7.40 |
| CutMix | 88.00 | 89.60 | 86.40 | 3.20 |
| AugMix | 90.00 | 89.20 | 90.80 | 1.60 |
| Cutout | 89.50 | 88.90 | 90.10 | 1.20 |
| Ours (External) | 91.10 | 89.80 | 92.40 | 2.60 |
| Ours (Internal) | 91.25 | 90.50 | 92.00 | 1.50 |
| Ours (Virtual) | 91.40 | 91.80 | 91.00 | 0.80 |

Table 15: **Performance comparison with image enhancement variants on UrbanCars (ViT-B/16).** We report accuracy on the original co-occurrence (I.D.), background shifted (BG), and Co-object shifted (Co-Obj) subsets, along with overall average accuracy.

| Method | Avg.↑ | I.D.↑ | BG↑ | Co-Obj↑ |
|---|---|---|---|---|
| CLIP | 63.07 | 82.00 | 37.20 | 70.00 |
| Mixup | 61.73 | 78.40 | 36.80 | 70.00 |
| CutMix | 62.00 | 77.20 | 38.40 | 70.40 |
| AugMix | 64.13 | 84.00 | 30.40 | 78.00 |
| Cutout | 62.80 | 80.00 | 36.40 | 72.00 |
| Ours (External) | 68.53 | 86.40 | 45.60 | 73.60 |
| Ours (Internal) | 68.27 | 88.00 | 45.60 | 71.20 |
| Ours (Virtual) | 71.87 | 89.60 | 48.00 | 78.00 |

Table 16 and 17 summarize the computational cost and efficiency analysis. Table 16 shows the FLOPs per image required for different methods, with the increments relative to the CLIP baseline in Waterbirds dataset. Our method's overhead is negligible compared to ALIA [28] and other augmentation-based methods, highlighting the advantages of representation-level reasoning.

Table 16: **Efficiency Comparison on Waterbirds across Different ViT Backbones.** GFLOPs per image is reported as increments relative to the CLIP baseline ("+" indicates additional GFLOPs). $M$ represents the number of counterfactual samples per image.

| Method | $M$ | ViT-B/32 | ViT-B/16 | ViT-L/14 | ViT-H/14 |
|---|---|---|---|---|---|
| CLIP (base) | – | 8.8214 | 35.1417 | 162.0866 | 334.6752 |
| ALIA (diffusion-based) | 7 | +1448068 | +1452343 | +1478801 | +1512448 |
| AugMix | 7 | +390.5 | +3995.7 | +28467 | +67880 |
| Ours (External) | 100 | +0.0027 | +0.0033 | +0.0053 | +0.0071 |
| Ours (Internal) | 100 | +0.0015 | +0.0023 | +0.0035 | +0.0043 |
| Ours (Virtual) | 100 | +0.0019 | +0.0025 | +0.0041 | +0.0055 |

Table 17 further evaluates the efficiency of our method as a function of the number of counterfactual samples ($M$) per image. As shown, while increasing $M$ generally improves performance, the additional computation cost remains very low, especially compared to conventional data augmentation methods. This demonstrates the scalability and efficiency of our approach, even for large datasets and high-throughput inference scenarios.

### E.3    Analysis of Module Parameters

We conduct ablation studies to analyze the sensitivity of our method to key hyperparameters under the virtual variant setting. Experiments are performed across four CLIP backbones, and results are reported in Fig. 11. We examine the following factors: the number of sampled counterfactual scenes $M$ used in the intervention module via Eq. (18); the hallucination suppression coefficient $\hat{\lambda}$ from

Table 17: **Efficiency with Varying Counterfactual Samples $M$ on Waterbirds (ViT-B/16).** FLOPs per image is reported as *increments relative to the CLIP baseline*.

| Method | $M = 50$ | $M = 100$ | $M = 200$ | $M = 300$ |
|---|---|---|---|---|
| Ours (External) | +0.0028 | +0.0033 | +0.0042 | +0.0051 |
| Ours (Internal) | +0.0018 | +0.0023 | +0.0032 | +0.0041 |
| Ours (Virtual) | +0.0020 | +0.0025 | +0.0034 | +0.0043 |

Eq. (15); and the predictive balance coefficient $\lambda$ controlling the weight of interaction terms in final classification in Eq. (19).

**Impact of the Number of Counterfactual Scenes $M$.** As shown in the left panels of each figure in Fig. 11, increasing $M$ initially improves performance across all backbones by introducing more diverse context embeddings, which enhances the effectiveness of the intervention. However, beyond a certain point, accuracy saturates and slightly declines. We attribute it to two factors: first, excessive samples may introduce redundant or noisy descriptions, especially in the virtual variant; second, the filtering mechanism becomes less effective as the number of samples increase, allowing lower-quality scenes to influence the final embedding. Notably, larger backbones such as ViT-L/14 and ViT-H/14 reach optimal performance with fewer samples, indicating that stronger models can generalize with smaller but higher-quality interventions.

**Impact of Predictive Balancing Coefficient $\lambda$.** The rightmost plots in Fig. 11 illustrate the influence of $\lambda$, which balances the predictive weight between the interaction in original image **i** and its intervention terms (see Eq. (19)). As $\lambda$ increases, performance improves initially, benefiting from richer imagined scenarios. However, overly high $\lambda$ may dilute the grounded signal in **i**, slightly hurting prediction reliability due to excessive dependence on synthetic embeddings.

**Impact of Hallucination Suppression Coefficient $\hat{\lambda}$.** As reflected in the center plots of Fig. 11, increasing $\hat{\lambda}$ leads to improved performance up to a moderate range (typically $\hat{\lambda} \in [0.5, 1.5]$). This coefficient controls the suppression of background-induced hallucination during the TDE computation (see Eq. (15)), by down-weighting contributions from background-only hallucination effects. Larger values of $\hat{\lambda}$ more aggressively remove spurious context effects, improving model reliability. However, if $\hat{\lambda}$ is set too high, it may overly penalize useful signals, causing slight drops in accuracy.

### E.4 Counterfactual Framework Pseudo-code

Algorithm 1 reflects our inference method combining representation-level counterfactual construction and total direct effect computation. Candidate object scores on token $p(y = c \mid v_j)$ are computed via sigmoid activation, while background likelihood is defined as $p_{\text{bg}}(j) = 1 - \max_c p(y = c \mid v_j)$. Tokens with $p_{\text{bg}}(j) > \tau$ are used to construct the background embedding $\mathcal{C}(z)$, while class-conditional object tokens satisfying $p(y = c_k \mid v_j) > \tau$ yield $\mathcal{C}(x_{c_k})$. We adopt thresholds for background and object embeddings.

The base prediction score is computed as Eq. (15), where the hallucinated contribution of background-only signals is controlled by suppression coefficient $\hat{\lambda}$. For each top-$K$ class on original prediction, we select $M$ counterfactual contexts $\{z_m\}$ that are jointly dissimilar to $\mathcal{C}(z)$ and $\mathcal{C}(x_c)$, ensuring semantic orthogonality of intervention. Counterfactual embeddings $\mathcal{C}(x_c, z_m)$ are generated by linearly combining $\mathcal{C}(x_c)$ and $f_i(z_m)$ with mixing ratio $\alpha$, and then used in Eq. (18) to simulate interventions. We note that although it aggregates over $M$ counterfactuals, practical implementation applies a filtering mechanism prior to sampling to avoid noisy or entangled contexts.

In Eq. (19), only top-$K$ predicted classes undergo counterfactual imagination and fusion, while others retain their base TDE scores, achieving robust calibration with minimal overhead. The entire inference is free of additional training, leveraging only frozen CLIP representations and batch-accessible or externally precomputed scene embeddings.

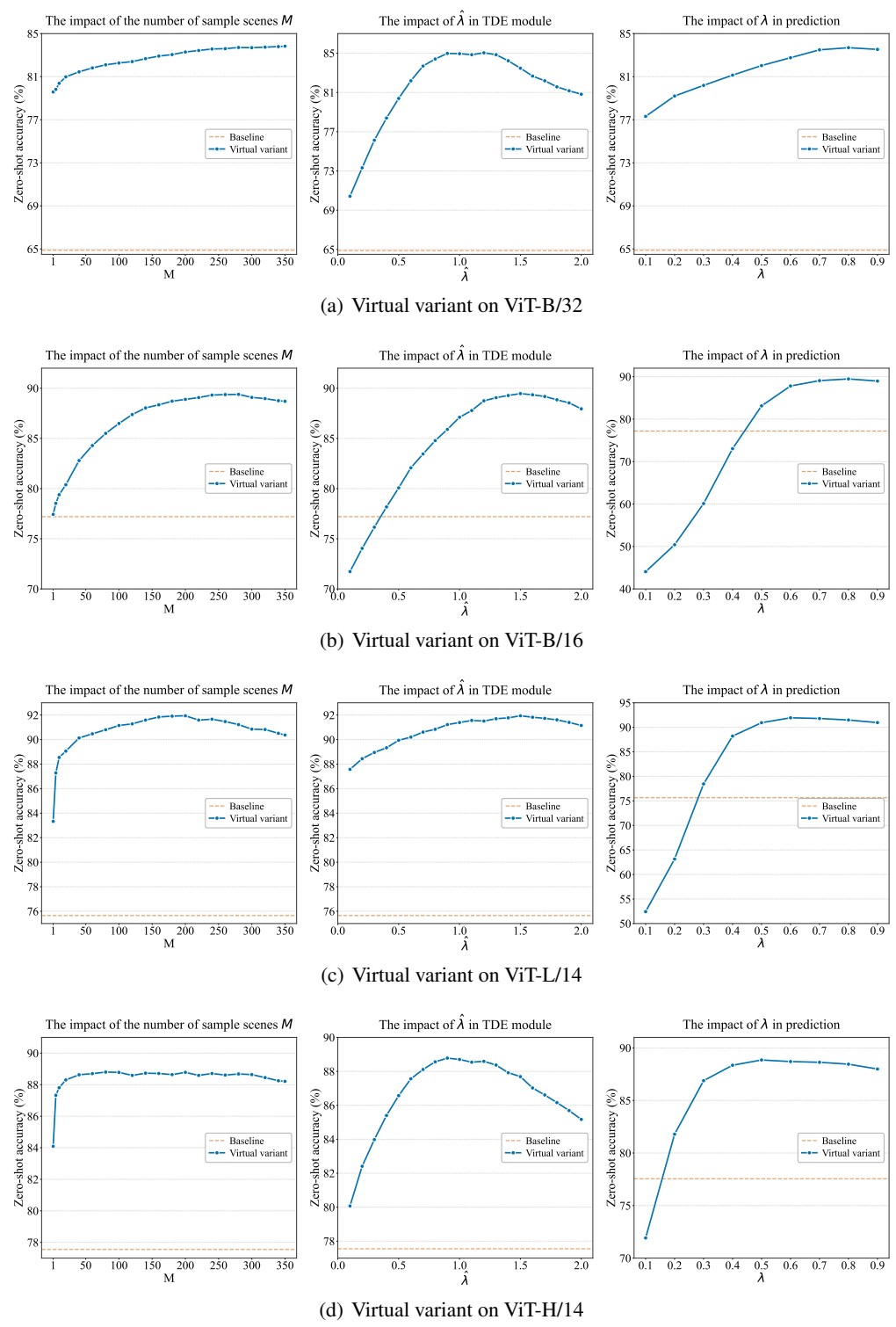

Figure 11: **Parameter analysis of the virtual variant on Waterbirds across different backbones**. Each panel shows the impact of (left) number of sampled counterfactual scenes $M$, (middle) hallucination suppression coefficient $\hat{\lambda}$ and (right) predictive balance coefficient $\hat{\lambda}$ on zero-shot accuracy.

**Algorithm 1:** INFERENCE WITH TDE AND REPRESENTATION-LEVEL COUNTERFAC-
TUAL CALIBRATION

**Input** : Token embeddings $\{v_j(\mathbf{i})\}_{j=1}^N$, text embeddings $\{f_t(T_c)\}_{c \in \mathcal{Y}}$;
parameters $\alpha, \lambda, \hat{\lambda}, \tau, K$; context pool $\mathcal{Z}_{\mathrm{src}} = f_i(\boldsymbol{z}_b)\}_{b=1}^B$
**Output** : Predicted label $\hat{y}$

**1. Estimate Main Effects from Image Tokens** :
Compute per-token class probabilities: $p(y = c \mid v_j) = \sigma(S(v_j, T_c))$ ;
Background score: $p_{\mathrm{bg}}(j) = 1 - \max_c p(y = c \mid v_j)$ ;
Background weights: $w_z(j) = \mathbb{I}(p_{\mathrm{bg}}(j) > \tau)$ ;
Candidate object weights (per class): $w_x^{(c)}(j) = \mathbb{I}(p(y = c \mid v_j) > \tau)$ ;

**2. Compute First-Order Embeddings** :
Background embedding:$\mathcal{C}(\boldsymbol{z}) = \mathrm{Normalize}\left(\sum_j w_z(j) \cdot v_j / \sum_j w_z(j)\right)$;
**foreach** *top-K class $c_k$* **do**
> Object embedding: $\mathcal{C}(\boldsymbol{x}_{c_k}) = \mathrm{Normalize}\left(\sum_j w_x^{(c_k)}(j) \cdot v_j / \sum_j w_x^{(c_k)}(j)\right)$

**3. Compute Base TDE score by Eq. (15)** :
$S_{\mathrm{img}} = S(f_i(\mathbf{i}), T_c), \quad S_{\mathrm{bg}} = S(\mathcal{C}(\boldsymbol{z}), T_c)$ ;
$\mathrm{TDE}_{\mathrm{base}}(c) = S_{\mathrm{img}} - \hat{\lambda} \cdot S_{\mathrm{bg}}$ ;

**4. Construct Representation-Level Counterfactuals by Eq. (11)** :
**foreach** *top-K class $c_k$* **do**
> Sample $M$ scenes $\{\boldsymbol{z}_m\} \subset \mathcal{Z}_{\mathrm{src}}$ ;
> Filter by dissimilarity to $\mathcal{C}(\boldsymbol{z})$ and $\mathcal{C}(\boldsymbol{x}_{c_k})$ ;
> Construct: $\mathcal{C}(\boldsymbol{x}_{c_k}, \boldsymbol{z}_m) = \alpha \cdot \mathcal{C}(\boldsymbol{x}_{c_k}) + (1 - \alpha) \cdot f_i(\boldsymbol{z}_m)$ ;

**5. Compute the intervention on object by Counterfactuals via Eq. (18)** :
$\mathrm{TDE}_{\mathrm{cf}}(c_k) = \frac{1}{M} \sum_{m=1}^M \left[ S(\mathcal{C}(\boldsymbol{x}_{c_k}, \boldsymbol{z}_m), T_{c_k}) - \hat{\lambda} \cdot S(f_i(\boldsymbol{z}_m), T_{c_k}) \right]$ ;

**6. Final Prediction Score by Eq. (19)** :
**foreach** *class $c \in \mathcal{Y}$* **do**
> **if** $c \in$ *top-K* **then**
>> $y(\mathbf{i}; c) = (1 - \lambda) \cdot \mathrm{TDE}_{\mathrm{base}}(c) + \lambda \cdot \mathrm{TDE}_{\mathrm{cf}}(c)$
> **else**
>> $y(\mathbf{i}; c) = \mathrm{TDE}_{\mathrm{base}}(c)$

**return** $\hat{y} = \arg\max_{c \in \mathcal{Y}} y(\boldsymbol{i}; c)$

# F  Limitation and Future Work

Our method operates entirely at the embedding level and relies on estimating object and background prototypes through linear aggregation. While effective, it may overlook certain spatial interactions to some extent. Additionally, performance gains depend partially on the diversity and representativeness of the available alternative contexts (external scenes, batch samples, or textual descriptions); thus, effectiveness may vary if suitable context alternatives are limited. Future work could extend our causal approach to multi-label and dynamic visual scenarios, refine nonlinear estimation methods for representation, and develop adaptive context-selection strategies to further enhance reliabilty and generalization.

