# OpenReview forum: "Representation-Level Counterfactual Calibration for Debiased Zero-Shot Recognition"
_NeurIPS.cc/2025/Conference — NeurIPS 2025 poster_

### Official Review · Reviewer_d8Ai · 2025-06-30

**Clarity:** 2
**Significance:** 2
**Originality:** 3
**Rating:** 4
**Confidence:** 4

**Summary:**

In this paper, the authors propose a theoretical causal approach to enhance zero-shot reliability of VL models (CLIP). Utilizing TDE, the authors emphasize the effective removal of context bias, and highlight the efficacy of the proposed method in terms of no need of retraining and heuristic prompt design, as well as practicality of the light-weight design.

**Questions:**

Q1. The reviewer thinks that the term “Hallucination” is wrongly defined in this paper. Assigning high scores for the albatross for the given sea image is misclassified results rather than the hallucinated results (many researchers may agree the term hallucination is normally used for generative models that generated persuasive responses not grounding on the factual information). The reviewer thinks that context bias is more proper terminology for the shortcuts.

Q2. The definition of semantic token effect seems too much simplified. (tbh sec 3.1 should be improved a lot - e.g., through line 160-167 the readers cannot have insights of the semantic designs for the implementation of the TDE formulation) The reviewer also agree of the importance of MSA for the feature extraction. But considering the complicated non-linearity in the transformer architecture, the reviewer concerns the role of MSA in the semantic effects (beyond the ablation results). Can the authors generalize this assumption on more general datasets or any other reference to support this claim?

**Ethical Concerns:**

["NO or VERY MINOR ethics concerns only"]

**Final Justification:**

Most of my concerns have been addressed during the rebuttal. I appreciate the clarifications and hope that the discussed points will be properly reflected in the potential final version.

**Limitations:**

The limitation or discussion section is not described in the currnet version.

Despite the impressive research topic and their theoretical revisit of CLIP models, the current method of counterfactual intervention is limited to certain benchmarks, which cannot support the efficacy in the more generalized cases (e.g., in-the-wild, ood, etc,.). The reviewer leans to negative for the current version and will discuss during the discussion period.

**Paper Formatting Concerns:**

N/A. Just minor comments on typos (line 29, 120)

**Quality:**

2

**Strengths And Weaknesses:**

The reviewer enjoyed to read this paper. Especially, the theoretical part of the feature decomposition and the origin of object-context bias in CLIP (sec 2) are quite impressive to follow. The authors theoretically present that the current SSL training (w/ infoNCE loss) inherently provoke entanglement of object-context in terms of information theory and empirical observations on toy examples support such claims.

Despite of the intriguing research direction, there are several points arguing about.

1. The experimental benchmarks are ideal playground to show the effectiveness of the proposed method due to the explicit definition of object-context bias in the datasets. The reviewer believes the essential strength of CLIP model is its powerful generalizability for the any given combination of the object-context, where the current method is limited show the effectiveness in such environments.
2. The proposed method requires context pool to alter the object-context pair, which is extracted from other datasets, other images within the batches or the generated features from text side. That is, the current approach relies on the predefined or closed range of context, which is limited to show the effectiveness under the open-ended and zero-shot cases.
3. In short, the formulation of TDE (eq. 14) is the score difference between the original image and (estimated) background-only image. Here, C(z) is the solely weighted summation of token embeddings that is mostly like background. But the improvement is more than 44%p in WB dataset, which is quite impressive and also doubtful. As aforementioned, the reviewer thinks that this may the results of the overfitting in the benchmarks (that includes only simple instances of single object and bg context).
4. The analyses of the proposed method is limited in this paper. The reviewer thinks that simply listing benchmark performance cannot deliver enough insights to potential readers (line 261-272).

---

> ### Author Rebuttal · Authors · 2025-07-31
>
> Thank you for your thoughtful review and encouraging feedback. We are glad that you considered our work “enjoyed reading, impressive theory, effective”. We are glad to address your questions.
>
> **Q1: Why do CLIP models still suffer from co-occurrence bias despite their strong zero-shot capabilities?**
>
> **A1:**  Thank you for raising this important point. While CLIP performs well on familiar object–context pairs, it also encodes object–context **shortcuts from its pretraining data** (e.g., LAION-2B), which limits its generalization to novel combinations. Our benchmark are designed to expose such biases, with worst-group accuracy dropping as **low as 34.5%** (Tab. 8). We believe addressing this issue is critical for the following reasons:
>
> -***Context bias is inherent in pre-trained VLMs.*** As shown in our PMI analysis (Fig. 2), LAION-2B shares gender–scene co-occurrence patterns with COCO and contains low-level artifacts like watermark–carton correlations [1]. These biases stem from the data itself and cannot be removed by simply scaling it, but can be addressed through targeted counterfactual intervention.
>
> -***Larger models amplify shortcut learning.*** Scaling to ViT-L/14 or ViT-H/14 does not resolve the issue. On the contrary, larger models tend to memorize and exploit spurious correlations more aggressively in ambiguous or occluded cases. This matches findings from Kaiming He et al. [2], which show stronger models are more sensitive to dataset-specific shortcuts, making bias mitigation even more crucial as models scale.
>
> -***Biases affect downstream systems.*** Many LVLMs, such as MiniGPT-4, LLaVA, use CLIP as their image encoder and inherit these hallucination patterns [3], leading to potential failures in real-world applications. Therefore, we believe CLIP remains a relevant and important testbed for studying and mitigating object–context bias.
>
> We will include these discussions in the revised version.
>
> $ $
>
> **Q2: Whether predefined context pools are limited in open-ended or zero-shot settings?**
>
> **A2:** Thank you for raising this insightful concern. We agree that **context pool diversity** is essential to our method. Our approach simulates interventions by imagining an object in alternative contexts at representation-level, which is conceptually similar to randomized trials. For these to be effective, we only require a diverse set of dissimilar contexts, which discourages reliance on pretraining shortcuts (see Fig. 11).
>
> In practice, the candidate pool is **not fixed or dataset-specific**. For structured datasets like Waterbirds, we can design targeted pools to match domain-specific scenes via a **priori**. For open-domain datasets like COCO, it makes sense to sample from Places365. Our method also **supports more larger open-world datasets** (e.g., Open Images, SUN397, ADE20K), which cover diverse real-world scenes without extra labeling costs. With our sampling strategy (lines 220–223), selecting from such pools remains efficient and lightweight.
>
> Moreover, the virtual variant enables text-guided counterfactual construction using LLMs like GPT-4, allowing for rich and diverse prompts beyond any limited scenes dataset. This makes **our method task-agnostic and extensible: once a diverse virtual context pool is built, it can be reused across tasks or deployed in zero-shot settings.**
>
> We will clarify this in the revision and emphasize that our method does not rely on a fixed context pool, but rather supports open-ended generalization through flexible sampling strategies.
>
> $ $
>
> **Q3: Is the high performance of the WB dataset a coincidence that TDE solves overfitting?**
>
> **A3:** Thank you for raising this question. We clarify that the large improvement on the Waterbirds dataset (over 44%) is not due to overfitting, but rather from effectively **reducing the strong object–context bias** that CLIP inherits from pretraining on LAION-2B. As shown in Tab. 8, CLIP performs well on common object–background pairs but fails on rare ones, which is precisely the issue our method targets.
>
> While some gain may come from using Places365 as an external pool which **aligns well with WB backgrounds**, we also observe strong performance with internal and virtual variants, confirming the method’s generalization across context sources.  Moreover, Our method **generalizes well across different CLIP backbones** (ViT-B/32 to H/14) **and datasets** (UrbanCars, COCO-GB, NICO), consistently improving worst-group accuracy. This suggests that it reliably mitigates hallucinations caused by contextual bias.
>
> Besides, the effectiveness is not due to a simple subtraction. It is derived from a **counterfactual optimal estimation framework** (Sec. 3.1), where we enable to compute the Total Direct Effect (TDE), using $\hat \lambda$ to suppress the confounding context-only pathway. This formulation, along with **multiple $C(z)$ variants simulating interventions** (Eq. 17), helps prevent context-driven misclassification, such as predicting “albatross” based only on the ocean in Fig. 1(b).
>
> We will revise the paper to clarify these distinctions and reinforce the broad applicability of our method beyond any single dataset.
>
> $ $
>
> **Q4: Clarification of the analytical experiments**
>
> **A4:** Thank you for pointing this out. We fully agree that in-depth analysis is crucial to help readers better understand our method. Due to space limitations in the main paper, many of our extended analyses were **placed in the appendix**.
>
> In particular, we provide additional experiments across four datasets in Appendix D, and introduce detailed pseudocode in Appendix C.3 to facilitate reproducibility. Moreover, Appendix E.2 offers a comprehensive  analysis of hyperparameters, and Appendix E.1 compares our method with 4–5 classical augmentation techniques by adapting them into counterfactual variants.
>
> Additionally, we plan to include **more comparison baselines and provide FLOPs analysis** (access to relevant tables in the responses to reviewers XSqm and 7obT), to further strengthen the evaluation. Finally, we will improve the main paper by explicitly pointing to relevant appendix sections, so readers can more easily access the extended analysis.
>
> $ $
>
> **Q5: Discussion on the use of term "hallucination"**
>
> **A5:** Thank you for raising this point regarding term. We clarify that, at the mechanism level, *object–context bias* refers to the learned co-occurrence shortcuts during training, while *hallucination* describes the inference-time outcome where the model makes overconfident predictions based on misleading contextual cues [4, 5].
>
> In our paper, “hallucination” refers to **cases where the model predicts a label even when the target object is absent**. For example, in Fig. 1(b), the model predicts “albatross” based solely on the ocean background. Alternatively, one can view it as CLIP **mistakenly retrieving or generating a caption** that strongly matches the background, but does not reflect the true content of the image.
>
> To avoid ambiguity, we will clarify it in the revised draft, avoiding the use of “hallucination” or replacing it with “context-induced misprediction”.
>
> $ $
>
> **Q6: Discussion on semantic-token direct effect**
>
> **A6:** Thank you for the insightful suggestion. We agree that Section 3.1 can be improved, and we will revise it to better explain how the semantic-token is captured. Specifically, we will clarify how the direct-effect decomposition helps assign distinct semantic roles to different tokens.
>
> **MSA dominates semantics.** Unlike MLPs which process each token independently, multi-head self-attention (MSA) allows tokens to exchange information, **enabling the model to gather semantic cues across spatial regions**. Prior works support this: Caron et al. [6] show that attention heads can recover segmentation structures even without supervision, and Naseer et al. [7] find that ViTs use long-range attention to model global semantics. Recent study [8] also shows that attention-guided token grouping transfers well across datasets and tasks. Our mean-ablation study on ImageNet (Tab. 5) further confirms that removing MSA significantly weakens semantic attribution.
>
> **Direct effect makes sense**. Although Transformers contain nonlinear components, our decomposition remains interpretable in CLIP, as the final prediction is computed by a **linear inner product** between image and text embeddings. This allows us to estimate each token’s semantic contribution to the final representation (see Eq. 6). Furthermore, ViTs include **residual connections** after each MSA and MLP block, allowing us to compare representations with and without specific modules (see Eq. 33), making direct-effect estimation possible even in nonlinear models.
>
> We will clarify these points in the revision and cite relevant work to support our method.
>
> **Reference:**
>
> [1] Li, Zhiheng, et al. "A whac-a-mole dilemma: Shortcuts come in multiples where mitigating one amplifies others." CVPR2023.
>
> [2] Zhuang Liu and Kaiming He. "A decade’s battle on dataset bias: Are we there yet?" ICLR2025.
>
> [3] Chen, Junzhe, et al. "Ict: Image-object cross-level trusted intervention for mitigating object hallucination in large vision-language models." CVPR 2025.
>
> [4] Liu, Yufang, et al. "Investigating and mitigating object hallucinations in pretrained vision-language (clip) models." EMNLP 2024.
>
> [5] Chen, Xuweiyi, et al. "Multi-object hallucination in vision language models." NIPS2024.
>
> [6] Caron et al., “Emerging Properties in Self-Supervised Vision Transformers,” ICCV 2021.
>
> [7] Naseer et al., “Intriguing Properties of Vision Transformers,” NeurIPS 2021.
>
> [8] Yossi Gandelsman, et al. "Interpreting CLIP’s image representation via text-based decomposition." ICML2024.

---

> > ### Comment · Reviewer_d8Ai · 2025-08-05
> >
> > Thank you for the detailed rebuttal.
> >
> > Most of my concerns have been addressed, and I appreciate the authors' effort to respond detailedly. In particular, my major concern was that CLIP's strongest appeal lies in its generalizability and zero-shot capabilities. However, the current experimental setup mainly focuses on benchmarks where object–context pairs are explicitly defined and may not fully reflect open-ended scenarios (tbh not persuasive enough in the response).
> >
> > As the authors noted in the rebuttal, context pool diversity plays an important role in simulating such open-world variation, and I think that extending evaluation to more unconstrained settings could be an important future direction (though I understand that it may be out of scope in current version.)
> >
> > Considering the contribution of thi paper, I am inclined to raise my score and lean toward BA.

---

> > > ### Author Response · Authors · 2025-08-05
> > > **Thank you for your feedback and suggestions**
> > >
> > > Thank you for your thoughtful comments. We sincerely appreciate your recognition of our efforts and your constructive suggestions.
> > >
> > > We fully agree that evaluating in more open-ended settings is an important direction. Current benchmarks (e.g., Waterbirds, COCO-GB, UrbanCars, NICO) are designed to test controlled object–context biases. While they cover only part of real-world scenarios, **they are useful for identifying the worst-case subgroups where models tend to struggle.** Evaluating on these subsets helps us better understand and validate the strengths of our approach.
> > >
> > > Our method is built to scale toward open-world use. Since it operates at the representation-level rather than generating at image-level, it allows **fast and large-scale creation of counterfactual examples**. In particular, our virtual variant mechanism **creates** context-aware embeddings **directly within the shared vision–language space**, making it flexible and efficient for broader settings.
> > >
> > > **Thank you again for your constructive feedback. Your support is truly valuable to us.**

---

### Official Review · Reviewer_pF3x · 2025-06-30

**Clarity:** 3
**Significance:** 4
**Originality:** 3
**Rating:** 5
**Confidence:** 3

**Summary:**

This work introduces a framework for improving the zero-shot classification performance of Vision-Language Models (VLMs) by reducing the impact of background features.

The CLS token used as image embedding is decomposed into multiple additive components which are then separated into background and object components by their CLIP score with respect to the class prompts.
The object component is then combined with new background components that can be obtained from external data, in-batch samples or “virtual” backgrounds from text embeddings, producing counterfactual embeddings which simulate the object on various other backgrounds.

The final prediction is based on a weighted sum of the Total Direct Effect (TDE, Eq. 14) of the original image and the average TDE of the simulated samples.

**Questions:**

Questions
1. The paragraph starting on line 117 seems a bit unclear to me. The authors supposedly want to compare a sample $(x, z)$ with another sample $(x’, z)$, but the inequality on line 123 rather resembles the difference in CLIP score of the sample $(x,z’)$ with respect to the classes $c_x$ and $c_{x’}$. My intuition is that the authors were referring to a $(x,z’)$ sample instead of  $(x’,z)$ on line 121 and that $z’$ is a background which favors the predictions of a class $c_{x’}$. Could the authors clarify whether there is simply a typo in this paragraph or I missed something else?

2. From Appendix B, lines 456-457: “All embeddings from dataset are assumed centered”. Do the authors themselves perform a centering of the embeddings in this regard?

Suggestions:
- I think that Algorithm 1 in the Appendix clearly illustrates the steps of the framework and that it could help a reader to more easily understand the proposed method. While there may be no room to add it in the main paper, I think that it would be a good idea to mention it somewhere in Section 3.
- The authors could consider also including the $\hat{\lambda}$ parameter in Eq. 14. It’s not necessarily a problem that it is only mentioned in the Appendix, but adding it shouldn’t overload the paper.

Typos (not an exhaustive list, I would recommend the authors to proofread the entire paper):
- Line 29 - o-occuring
- Line 88 - the bias(es?) are enhanced
- Line 95 - will inherently entangle**s**
- Line 123 - I think that $f_t$ should be used for the embeddings of texts instead of $f_i$
- Line 159 - shouldn’t the attention weights sum up to 1 instead of 0?
- Line 184 - missing blank space between the dot and the first word of the next sentence
- Line 201 - sentence starting in lower case - “we thus update…”
- Line 453 - I think the authors meant to use $X \in \mathcal{X}$ instead of $X \in \mathcal {Y}$, as in Line 67
- Line 457 - Shouldn't the second expectancy be of $f_t(T)$?

**Ethical Concerns:**

["NO or VERY MINOR ethics concerns only"]

**Final Justification:**

This paper introduces a method for synthesizing counterfactuals directly in the embedding space of CLIP models. As shown in the rebuttal, this approach is much more computationally efficient than using input-level editing with diffusion models.

The counterfactuals are successfully used to reduce the dependency of CLIP-based zero-shot classifiers on background features over four datasets.

The authors have addressed the concerns of the reviewers and clarified some parts of their work in the rebuttal.

**Limitations:**

The authors discuss the limitations of their framework in Appendix F.

**Paper Formatting Concerns:**

No formatting issues were noticed.

**Quality:**

3

**Strengths And Weaknesses:**

Strengths:
- The proposed methodology is sound and it surpasses prior works in the considered setups
- The framework is validated on multiple datasets and ablations of its components (Table 4) or hyper-parameters (Appendix E.2) are provided
- The method does not have heavy hardware requirements
- Implementation details are properly provided, the code itself is also shared

Weaknesses:
- As noted by the authors in the Limitations and Future work section, the end performance depends upon the available backgrounds for synthesizing the counterfactual embeddings. While an ablation is performed on the number of backgrounds (scenes) used (Appendix E.2, Fig. 11), it is unclear what the results would look like if the pool of background scenes lacks in diversity (for any setting among external, internal or virtual). For the external setting of the Waterbirds dataset the authors use scenes from the Places-365 dataset, which is the very source of background pictures used in creating the Waterbirds dataset. To some extent, this could explain why the highest performance gains are obtained on this dataset. A simple test in this regard would be to use either only water-related backgrounds or land-related backgrounds in the virtual setting for the Waterbirds datasets and see how that affects the group-wise accuracies. This might end up similarly to having a low value for M (as in Fig. 11), so implementing it is not a must.
- Some results are harder to read out as the dataset on which they are obtained is not explicitly mentioned in the caption of the table or figure, so the reader must first find the corresponding paragraph where they are referenced - Tables 1, 4, 11, 12, Figure 11.

---

> ### Author Rebuttal · Authors · 2025-07-31
>
> Thanks for your appreciation of our paper. We are glad that you considered our work “sound, effective, easy to follow”. We are glad to answer all your questions.
>
> **Q1: Discussion on the impact of context diversity**
>
> **A1:** Thank you for the insightful comment on the diversity of the background pool for synthesizing counterfactual embeddings. We fully agree that **the quality, especially the diversity, of counterfactual embeddings is more critical than their quantity**. As discussed in lines 94–97 and analyzed in Fig. 11, a diverse candidate pool better approximates ideal interventions and more effectively mitigates biases embedded during training.
>
> Our sampling strategy (lines 220–223) is designed to achieve scene diversity and semantic plausibility. The external and internal variants sample from Places365 and in-batch sources, respectively, helping align the synthesized embeddings with real-world object–context co-occurrences. Additionally, we introduce a virtual variant, where counterfactual embeddings are constructed directly in the shared vision–language space via text. Leveraging recent advances in LLMs such as GPT-4, **we can efficiently generate rich and diverse scene prompts at scale, enabling low-cost counterfactual construction.**
>
> This approach also offers a practical benefit: it enables **task-agnostic deployment**. Once a diverse virtual pool is built, the same intervention strategy can be reused across tasks without additional manual effort.
>
> We will clarify these points in the revised draft, and we thank the reviewer again for emphasizing this important aspect.
>
> $ $
>
> **Q2: Clarifying table and figure representation**
>
> **A2:** Thank you for pointing this out. We will revise the layout and captions to make the results easier to read. Specifically, we will explicitly indicate the corresponding dataset in the captions of Tables 1, 4, 11, 12, and Fig. 11, and **place each table or figure closer to its first mention in the text**. We will also include brief descriptions of the metrics and datasets in each caption to avoid confusion and reduce back-and-forth reading. These changes will be reflected in the revised version to enhance readability and accessibility.
>
> $ $
>
> **Q3: Explanation of typos on lines 121-124**
>
> **A3:** Thank you for pointing out this confusion. You are right, and there is indeed a typo in lines 121–124. Our intention was to **illustrate how a novel background $z'$ can reverse the prediction for the same object $x$,** especially considering the co-occurrence pairs $(x,z)$ and $(x',z')$ observed during training. Concretely, consider a novel pair $(x,z')$ that rarely (or never) co-occurred during training, and thus has a low interaction term $r_i$. Define the decision margin as:
> $$
> \delta = S(\mathbf i_{x,z'},T_{c_x})-S(\mathbf i_{x,z'},T_{c_{x'}})
> $$
> where $c_x$ is the true class of $x$ and $c_{x'}$ is a confusable alternative.
>
> Then, using the decomposition in Eq. 4, this margin can be rewritten as:
> $$
> \delta =(\langle e_i(x), f_i(T_{c_x}) \rangle - \langle e_i(x), f_i(T_{c_{x'}}) \rangle)
> \;-\;
> (\langle e_i(z'), f_i(T_{c_{x'}}) \rangle - \langle e_i(z'), f_i(T_{c_x}) \rangle)
> $$
> This expression shows that if the contextual interaction effect (inherited from the co-occurrence pairs $(x',z')$) dominates the object evidence (the first term), then $\delta<0$ and the model incorrectly favors $c_{x'}$. This explains the failure case shown in Fig. 1(a), and is the underlying mechanism for the hallucinated scores visualized in Fig. 1(b).
>
> We will correct the notation and clarify the intent in the final version.
>
> $ $
>
> **Q4: Clarification on whether the embeddings are centered**
>
> **A4:** Thank you for raising this point. Yes, **we explicitly apply mean-centering to the embeddings.** Before downstream operations, such as CLIP inner product scoring or TDE estimation, we subtract the mean embedding (computed over the full dataset or mini-batch) from each vector. This ensures that the theoretical assumption in Appendix B (lines 456–457) is faithfully reflected in our actual implementation. We will publish the algorithm code in the future.
>
> $ $
>
> **Q5: Suggestions on clarity, presentation, and typos**
>
> **A5:** Thank you for the detailed and constructive suggestions. We agree that Algorithm 1 provides a clear summary of our method, and we will explicitly reference it in Section 3 to help readers navigate to the full procedure in Appendix C.3. We also appreciate the recommendation to introduce the hyperparameter $\hat{\lambda}$ earlier, and will include a brief explanation alongside Eq. 14 in the main text.
>
> Additionally, we are grateful for the careful review of typos, inconsistent notations, and formatting issues (e.g., notation for $f_t$, attention weights, expectation symbols). We will thoroughly proofread the paper and correct all potential issues to improve overall clarity and precision in the final version.

---

> > ### Comment · Reviewer_pF3x · 2025-08-01
> > **Side note on Q4 and an additional idea**
> >
> > I thank the authors for clarifying all the raised questions.
> >
> > **Side note on Q4.**
> >
> > Based on the authors’ answer I assume that the text embeddings are also centered by subtracting the mean over a larger body of text embeddings. This centering of embeddings is used in \[1\] to address the modality gap \[2\] in VLMs such as CLIP, a phenomenon that I did not notice being discussed in the paper and one that I also forgot to bring up in the initial review.
> >
> > While the authors already had their theoretical reasons for applying this centering, I also wanted to take this chance to mention the underlying modality gap that the centering of embeddings seemingly addresses.
> >
> > **Qualitative analysis of counterfactuals**
> >
> > After reading the other reviewers’ comments and the authors’ responses I thought of a way to qualitatively show that the counterfactual embeddings do indeed simulate the object on a different background \- an idea that lies at the core of the paper. I think that **it is not mandatory to have this experiment now** and that the authors can leave it for a future revision, but it could further strengthen the current submission.
> >
> > Reviewer d8Ai mentioned that “**simply listing benchmark performance cannot deliver enough insights to potential readers**”, and I do agree with him. Some works start from reasonable ideas of what a model could do to achieve better results, and then propose modifications that could intuitively lead to the desired behaviour or properties.  However, as deep networks are pretty much black boxes, we can not always assert that they truly learn the desired behaviour solely based on an increased benchmark performance, as that could also have been achieved for different reasons.
> >
> > The authors have provided attention maps based on $\\mathcal{C}(x)$, which suggest that the obtained $\\mathcal{C}(x)$ better captures the object of interest than the basic CLIP embedding, aligning with the intuition of what $\\mathcal{C}(x)$ represents. However, the fact that a synthesized counterfactual embedding $\\mathcal{C}(x, z’)$ truly corresponds to the desired object $x$ and background $z’$ may not entirely be backed by the provided experiments or qualitative examples. Validating this may help convince a reader of the method’s value.
> >
> > As the authors have mentioned in their responses, VLMs such as miniGPT-4 or LLaVA use CLIP as an image encoder. While those two seem to use all image patch tokens, ClipCap only uses the CLS embedding from CLIP to generate a caption for the image. It is thus possible to feed a counterfactual $\\mathcal{C}(x, z’)$ built from 2 different images $(x,z)$ and $(x’, z’)$ to ClipCap and observe whether it mentions the known object $x$ and background $z’$ or not.
> >
> > **References**
> >
> > \[1\] Bhalla et al. “Interpreting CLIP with Sparse Linear Concept Embeddings (SpLiCE)” NeurIPS 2024
> >
> > \[2\] Liang et al. “Mind the Gap: Understanding the Modality Gap in Multi-modal Contrastive Representation Learning” NeurIPS 2022

---

> > > ### Author Response · Authors · 2025-08-02
> > > **Discussion on embedding centering impact and counterfactuals' qualitative analysis**
> > >
> > > Thank you for the thoughtful follow-up and constructive suggestions.
> > >
> > > We appreciate your note regarding ***embedding centering***. While our initial motivation was to **satisfy the theoretical assumptions** stated in Appendix B (lines 456–457) and to **ensure a consistent and interpretable process** when synthesizing counterfactual embeddings from object and context representations, we now recognize, thanks to your observations in [1] and [2], that this step may also contribute to reducing the modality gap in VLMs and a extra ablation test can prove it. We will clarify this point in the revised version.
> > >
> > > $ $
> > >
> > > Moreover, we find your idea of ***qualitative validation*** particularly inspiring and a further complement to our current analysis, especially **in response to Reviewer d8Ai’s concern**. As you pointed out, our framework synthesizes counterfactual embeddings $\mathcal{C}(x, z')$ to simulate the same object in alternative contexts, aligning with the goal of counterfactual reasoning. **While this version of the paper focuses on group-level fairness and scores, we agree that checking whether $\mathcal{C}(x, z')$ retains the target semantics is crucial.**
> > >
> > > Using models like ClipCap or BCLIP to generate captions from the synthesized $\mathcal{C}(x, z')$ provides an intuitive way to validate the reliability. This aligns well with the intended design of our virtual variant, which leverages CLIP’s shared semantic space. **To some extent, it could also help guide the tuning of hyperparameters $\alpha$ or inspire more effective strategies for synthesizing counterfactuals.** We plan to include some qualitative examples in the revised version and our follow-up work.
> > >
> > > $ $
> > >
> > > Thank you again for your constructive suggestions. If you have any further questions or thoughts, please let us know and your insights are truly valuable to us.
> > >
> > > **References:**
> > >
> > > [1] Bhalla et al. “Interpreting CLIP with Sparse Linear Concept Embeddings (SpLiCE)” NeurIPS 2024
> > >
> > > [2] Liang et al. “Mind the Gap: Understanding the Modality Gap in Multi-modal Contrastive Representation Learning” NeurIPS 2022

---

### Official Review · Reviewer_7obT · 2025-07-01

**Clarity:** 2
**Significance:** 2
**Originality:** 3
**Rating:** 5
**Confidence:** 2

**Summary:**

This paper handles the zero-shot recognition problem from the causal view. It tackles the out-of-distribution problem during the inference time and formulates it as a causal inference question: Would the prediction remain if the object appeared in a different environment? The paper also proposes a counterfactual estimation method to operate directly at the representation level and achieves state-of-the-art zero-shot recognition performance across multiple context-sensitive datasets.

**Questions:**

The authors discuss the inference speed in the cost section, that is good. But I am also curious about what is the efficiency of the proposed method compared with standard CLIP in terms of flops?

I wonder if for stronger models with larger compacity, whether the proposed approach could work more and whether it can be generalized and whether if there is scaling law applies here. From Figure 1 (a), it looks like ViT-B/32 and ViT-H/14 drop more performance from the best to the worst group.

Is the method trainable (applied to the training paradigms)? Normally, when more data gets involved in training, the OOD problem can be more greatly mitigated.

**Ethical Concerns:**

["NO or VERY MINOR ethics concerns only"]

**Final Justification:**

Thank the authors for their detailed response. The clarifications regarding the choice of CLIP as the base model, the motivation from a causal perspective, and the explanation of computational efficiency address some of my concerns. The additional quantitative results and implementation details also address many of the concerns raised in the initial review. I will raise my rating.

**Limitations:**

Yes.

**Paper Formatting Concerns:**

I would suggest adding an implementation details section in the main paper to understand how exactly the method works.

**Quality:**

3

**Strengths And Weaknesses:**

Strengths:

- The problem formulation in this paper is interesting and novel. It gives a proper explanation and solution by formulating the problem from the causal perspective and model co-occurrence biases in terms of beneficial interactions and potential hallucinations, and explain the sensitivity of prompt-based strategies.

- There is solid theoretical explanations and proofs in this paper.


Weaknesses:
- The motivation of this paper seems to be weak, the author claims " leading models to rely on context rather than the true causal semantics, and ultimately degrading their zero-shot recognition performance, particularly when the test-time context diverges from familiar training distribution. The issue is especially pronounced in CLIP". But recent models including encoder like Dino, PE, etc, can do zero shot recognition very well even on unseen context.
- The experiments are only based on the CLIP model, which is kind of toy and outdated. There are many more advanced models such as PE, BLIP, SigLIP, etc, while the paper does not try.
- The datasets covered in this paper only include Waterbirds, UrbanCars, COCO-GB, and NICO. While there are a lot of interesting datasets outside there for zero-shot recognition such as Winoground etc, which are not tested.
- Although the paper did a detour to introduce the method from the causal perspective and give many theorems and proofs, but the actual implementation of the method is unclear, even from the Appendix it is still hard to reproduce the method and not sure if the method can be generalized.
- From Figure 4 and the method introduction, it looks like the method can add a lot of computational complexity. The authors should also compare the efficiency of the proposed method with other baselines like flops, memory metrics.

---

> ### Author Rebuttal · Authors · 2025-07-31
>
> Thanks for your constructive comments. We are glad that you considered our work “novel, well-structured theoretical, interesting” and we will answer all your questions.
>
> **Q1: Motivation and reason for selecting CLIP**
>
> **A1:** Thank you for raising these valuable concerns. We are glad to clarify our motivation in the revised version.
>
> **-*Stronger encoders do not eliminate co-occurrence bias*:** While recent backbones (e.g., DINO, PE) improve average, they still rely on object-context shortcuts **in challenging or occluded scenarios** [1]. Dataset classification studies [2] interestingly show that **larger pre-trained models amplifies such easy inter‑domain cues and passes them to downstream tasks** [1]. PMI in our paper confirms that LAION‑2B (CLIP’s pretraining dataset) shares the same gender‑scene patterns as COCO, meanwhile Li et al. [3] reveal watermark–carton co‑occurrences that skew predictions. Thus, simply switching to larger encoders or datasets does not remove co‑occurrence bias and it continues to threaten both safety and fairness.
>
> **-*Lightweight way to address shortcut reliance, not encoder competition*:** Our goal is to provide a lightweight, inference-time plugin to mitigate hallucinations, without the high cost of generative editing. We adopt CLIP-style VLMs as our base models because CLIP serves as a strong and widely used baseline, and its contrastive pretraining reveals the object–context shortcuts analyzed in Section 2.
>
> Moreover, our method is architecture‑agnostic: built on attention‑based direct‑effect decomposition and TDE, it can be transferred to stronger encoders. We provide initial evidence across four CLIP backbones with increasing capacity, demonstrating its compatibility (Tab.1, 8-12). **Starting with the CLIP allows a clean theoretical exposition and avoids interference from extra modules in newer models**.
>
> $ $
>
> **Q2: Additional benchmark**
>
> **A2:** Thank you for this suggestion. We deliberately chose Waterbirds, UrbanCars, COCO‑GB, and NICO because each dataset **explicitly encodes object‑context shortcuts** (e.g. backgrounds, gender cues, co‑occurring objects, etc.) that **let us quantify worst‑group accuracy**, the primary metric for our debiasing method. As shown in Fig. 1 and Tab. 8, worst‑group accuracy reveals a model’s true bias sensitivity: even strong zero‑shot CLIP variants score only ~40 % on this metric across all four architectures.
>
> Datasets such as Winoground are designed for fine‑grained text–image reasoning. They focus on text-generated hallucinations or caption mismatches and also do not report group‑wise accuracy. Hence they are better suited to evaluating LVLMs hallucinations when CLIP is used merely as an image encoder  component [1]. While that setting is not our current focus, our method can be applied to MiniGPT‑4, LLaVA, etc. methods on these datasets. In future work we plan a broader evaluation on benchmarks like Winoground to complement the object‑context bias tests.
>
> $ $
>
> **Q3: Algorithm implementation clarity**
>
> **A3:** We are glad to explain any ambiguity. The complete implementation can already be reconstructed from three parts of the paper:
>
> The **end-to-end inference pipeline is detailed in *Algorithm 1*** (Appendix C.3, p. 29):
>
> (i) extracting main-effect embeddings (Eq. 6 and 8) → (ii) constructing counterfactual samples(Eq. 15 and 16)→ (iii) fusing scores (Eq. 14, 17, and 18).
>
> Additionally, we provide ready-to-use YAML configs for each dataset (Waterbirds, UrbanCars, COCO-GB, NICO) in **the original supplementary README.md**, including the sampling script and full hyperparameter settings. In the revised version, we will reference Algorithm 1 more clearly in the main paper, expand more hyperparameter details, and release a public repo for reproducibility.
>
> $ $
>
> **Q4: Quantitative Analysis of Efficiency**
>
> **A4:** We appreciate the reviewer’s concern. While our method introduces additional inference steps, we emphasize that all operations are performed in the embedding space, with **no image re-encoding or pixel-level augmentation** required.
>
> We agree that quantitative metrics are essential for evaluating overhead. In the revised version, we will add some experiments, including comparing to ALIA [4], a language-guided generative baseline. By default, FLOPs measures the total number of floating-point operations, assuming pre-computed text embeddings. **Values in table are reported as FLOPs *increments* relative to CLIP baseline.**
>
> | Method(GFLOPs per image) |      | ViT-B/32 | ViT-B/16 | ViT-L/14  | ViT-H/14  |
> | ------------------------ | ---- | -------- | -------- | --------- | --------- |
> | CLIP                     |      | =8.8214  | =35.1417 | =162.0866 | =334.6752 |
> | ALIA  ($M=7$)            |      | +1448068 | +1452343 | +1478801  | +1512448  |
> | AugMix ($M=7$)           |      | +390.5   | +3995.7  | +28467    | +67880    |
> | External ($M=100$)       |      | +0.0027  | +0.0033  | +0.0053   | +0.0071   |
> | Internal ($M=100$)       |      | +0.0015  | +0.0023  | +0.0035   | +0.0043   |
> | Virtual ($M=100$)        |      | +0.0019  | +0.0025  | +0.0041   | +0.0055   |
>
> Table Note: Efficiency on Waterbirds across four different backbones during inference-time. $M$ is the number of counterfactual samples per image in Eq. 15.
>
> | Method(GFLOPs per image) | $M=50$  | $M=100$ | $M=200$ | $M=300$ |
> | ------------------------ | ------- | ------- | ------- | ------- |
> | Ours(External)           | +0.0028 | +0.0033 | +0.0042 | +0.0051 |
> | Ours(Internal)           | +0.0018 | +0.0023 | +0.0032 | +0.0041 |
> | Ours(Virtual)            | +0.0020 | +0.0025 | +0.0034 | +0.0043 |
>
> Table Note: Efficiency on ViT-B/16 varies from the number of counterfactual samples $M$ on Waterbirds.
>
> As shown above, **the computational introduced by our method is minimal, especially compared with generative approaches.** CLIP performs a single forward pass through a frozen encoder, and our method preserves this efficiency: it constructs counterfactual embeddings using closed-form computations (Eq. 9, 17) over precomputed token features, followed by lightweight TDE fusion (Eq. 18). **All operations involve only matrix multiplications and element-wise computations, without any backpropagation or additional network training.** Moreover, since the context candidate pool can be prepared offline, our method remains highly efficient in large-scale data deployment..
>
> $ $
>
> **Q5: Discussion on scaling law in our work**
>
> **A5:** Thank you for this insightful question. Our results on Waterbirds (Fig. 1(a) and Appendix Tab. 8) closely mirror those reported by Kaiming He et al. [2]. Across four CLIP backbones' results, we draw two key conclusions:
>
> -***Small models (B/32, B/16)*:**  Their limited ability under-represents the object itself, so worst-group errors stem primarily from **insufficient object representation**. Increasing capacity or resolution directly boosts both average and worst-group accuracy.
>
> -***Large models (L/14, H/14)*:** For “easy” high-resolution images, object cues quickly saturate. Extra capacity then tends to **amplify memorized object–context shortcuts** in ambiguous or occluded cases, leading the model to “guess” from background more aggressively, precisely the hallucination our method mentions.
>
> This pattern mirrors tests in [2]: even a tiny 7K-parameter ConvNeXt can achieve 72.4% on dataset-ID classification, rising to 85% when scaled to 87M parameters. Intuitively, classifying dataset splits should be randomly, but larger models systematically learn dataset-specific shortcuts, producing stronger hallucinations downstream [1].
>
> Hence, **increasing backbone scale without bias control actually reinforce shortcut learning**. In contrast, our representation-level counterfactual plugin pairs integrates with any model size to preserve object recognition while suppressing context-driven hallucinations,enabling safer and more reliable deployment. We plan to clarify these scaling effects in the revised version.
>
> $ $
>
> **Q6: Discussion on training paradigms**
>
> **A6:** This is a very insightful question. Although current design is inference-only, the counterfactual mechanism is **compatible with training paradigms**. The key idea which modifies object–context combinations via counterfactual synthesis, can be naturally integrated as a **data augmentation or regularization strategy** during training.
>
> More broadly, we agree that incorporating more data often enhances generalization. However, **data scaling alone is insufficient**—large-scale datasets tend to contain **spurious correlations** that models exploit as shortcuts. Our counterfactual approach provides a **targeted and controllable intervention** to address it directly.
>
> In particular, the **virtual variant** constructs semantically grounded counterfactual embeddings using text prompts in the vision–language space. This enables the generation of **lightweight synthetic supervision signals** without retraining or external annotation cost. It also makes our approach suitable for scalable training-time integration in the future, offering a promising path to debias models.
>
> **Reference:**
>
> [1] Chen, Junzhe, et al. "Ict: Image-object cross-level trusted intervention for mitigating object hallucination in large vision-language models." CVPR 2025.
>
> [2] Zhuang Liu and Kaiming He. "A decade’s battle on dataset bias: Are we there yet?" ICLR2025.
>
> [3] Li, Zhiheng, et al. "A whac-a-mole dilemma: Shortcuts come in multiples where mitigating one amplifies others." CVPR2023.
>
> [4] Dunlap et al., “Diversify your vision datasets with automatic diffusion-based augmentation,” NeurIPS2023.
>
> [5] Yossi Gandelsman, et al. "Interpreting CLIP’s image representation via text-based decomposition." ICML2024.

---

### Official Review · Reviewer_XSqm · 2025-07-02

**Clarity:** 3
**Significance:** 2
**Originality:** 2
**Rating:** 4
**Confidence:** 3

**Summary:**

Deep learning models, including vision-language models, often tend to overfit on their training sets, which may include background, texture, and shape information. This paper approaches the issue by framing it as a causal inference problem, raising an important question: If an object appears in a different environment, will the prediction still hold true?
The authors use a CLIP model to examine its representation space. They estimate the total direct effect and subtract the activation related to the background only, which allows them to preserve object-content interactions. Notably, they propose a method to enhance performance in a zero-shot manner, without the need for prompt design or retraining. They also assert that their framework represents a lightweight, counterfactual-based approach. Furthermore, the additional theory and experiments, which include analyses of module parameters and comparisons to classical augmentation techniques, strengthen their proposal.

**Questions:**

- Regarding the implementation details, how many samples are considered at test time to obtain the counterfactual embeddings?
- As an additional relevant baseline, I suggest that the authors include simple data augmentations as described in [2] for an input image and report the accuracy. This can be considered a first step in addressing the generalization problem at inference.
- This recommendation is in addition to the baseline comparisons outlined in Appendix E1, which include mixup, cutmix, augmix, and cutout.
- The performance on the CoCo-GB dataset appears to be incremental and not the best. Can the authors provide a justification for this performance drop?
- The authors currently use multiple predictions (ensemble of predictions) over multiple (generated) inputs. I am curious to see the performance of 'multiple models' (ensemble of models) on single input as a comparison. Needless to say, these needs multiple models compared to their approach

**Ethical Concerns:**

["NO or VERY MINOR ethics concerns only"]

**Final Justification:**

The authors have responded well to my questions during the rebuttal. I am in favor of acceptance, hence I will keep my original score.
The original submission lacked discussions about focusing on contextual bias, additional classical computer vision augmentation comparisons, ALIA [4], and its implementation, adjusting for better zero-shot performance. The reviewers have now answered these questions in a satisfactory manner.

**Limitations:**

Yes

**Quality:**

3

**Strengths And Weaknesses:**

Strengths
- It is interesting to note that counterfactual embeddings are obtained without relying on any external models.
- Overall, the paper is well-written, and the authors' contribution is clear.
- The inference costs are noteworthy and minimal, despite multiple counterfactual embeddings
- The experiments and theoretical aspects are designed and articulated clearly, effectively complementing the paper's approach.

Weaknesses
- Similar work by Sauer et al. (2021) [27] utilizes multiple GANs, where the object is decomposed into texture, shape, and background using external networks. This raises the question of why the authors are solely decomposing the object into its background. To what extent does overfitting or ambiguity in inference occur solely because of the background?
- The proposed method enforces consistency in predictions across diverse scenarios, which is similar to the concept presented in Test-Time Augmentation papers, such as Memo. I recommend that the authors cite this work.

References:
[1] Zhang, Marvin, Sergey Levine, and Chelsea Finn. "Memo: Test time robustness via adaptation and augmentation." NeurIPS 2022

---

> ### Author Rebuttal · Authors · 2025-07-31
>
> Thanks for your appreciation of our paper. We are glad that you considered our work “well-written, effective, interesting”. We are glad to answer all your questions.
>
> **Q1:** **Comparison to [1]  and why focus mainly on contextual bias**
>
> **A1:** Thanks for the insightful comment. While Sauer et al. [1] utilize generative models to disentangle different components, our method focuses specifically on addressing contextual bias considering in three aspects:
>
> **- *Empirical evidence that context drives most failures*:** Well-trained models easily capture low-order, “easy-to-learn” semantics, while high-order cues are harder to generalize [2]. In our error analysis on Waterbirds and COCO-GB, we found that most misclassifications arise from small, ambiguous, or obstructed object, which leads the model **tend to over-rely on background**. After all, we wouldn’t want a system that sees a kitchen and always predicts woman, or sees shrubbery and always predicts landbirds.
>
> **- *No external generative networks or fine-tuning*:** Disentangling texture and shape requires training or fine-tuning separate encoder-decoder networks. In contrast, we operate directly at inference time on a frozen CLIP model, using a closed-form solution (Eq. (9), Appendix B.7) to estimate object and background direct effects. This **avoids** the need for **additional parameters and training**.
>
> **- *Better zero-shot generalization*:** Introducing dedicated texture and shape models may **anchors the model to its specific data domains** (e.g., bird textures in Waterbirds), which **limits transfer to novel** categories or instances (e.g., humans appearances in COCO-GB). Since context is more readily decoupled from object identity than low-level appearance, our strategy better supports zero-shot recognition performance.
>
> We will include these discussions in the revised version.
>
> $ $
>
> **Q2: Supplemental citation [3]**
>
> **A2:** Thank you for suggesting the Memo [3]. We have added this citation in Section A and clarified that, differing from pixel-level TTA methods (which adjusts model more robust under augmentations), our framework **operates in the semantic space of frozen CLIP**. It decomposes and subtracts background effects and synthesizes counterfactual embeddings. This causal, representation-level approach directly addresses the issue of object-context confusion and controls hallucination effects caused by context.
>
> $ $
>
> **Q3: Experimental details on sample size for constructing counterfactual embeddings**
>
> **A3:** Thank you for your interest in experimental details. The full parameter settings, including sample size selection, are provided in the README.md of our original supplementary materials. We also promise to release the code in the future.
>
> Regarding the number of counterfactual samples generated, we conducted a parameter analysis in Fig. 11. For instance, the ViT backbones with increasing parameter sizes in Waterbirds , the virtual variant method sampled 270, 270, 190, and 90 from pool, as shown in Tab. 1 (sampling rules can be found in lines 220-223). **The scale of the candidate pool (400) can be adjusted by the experiment designer offline. Larger and more diverse candidate pools may result in fewer size.**
>
> $ $
>
> **Q4: Additional baseline information and clear baseline comparisons**
>
> **A4:** Thank you for the helpful suggestion. While the information citation "[2]" may have been unintentionally omitted by reviewer, we agree with the intention of introducing more augmentation-based baselines. Beyond Mixup, CutMix, AugMix, and Cutout **shown in Appendix E.1 (Tab. 13, 14)**, we further include **ALIA** [4], a diffusion-based method that performs language-guided image editing. We adapt ALIA into a counterfactual variant (see lines 773–777) and report its results. However, it incurs much higher computation cost compared to our lightweight representation-level approach.
>
> | Method         | Avg. ↑ | Worst. ↑ | FLOPs per image ↓ |
> | -------------- | ------- | ---------- | --------------- |
> | CLIP           | 77.20   | 50.93      | 8.8214 G        |
> | **ALIA** [4]   | 78.61   | 55.18      | +1452343 G      |
> | Ours(External) | 86.40   | 70.87      | +0.0033 G       |
> | Ours(Internal) | 85.43   | 82.71      | +0.0023 G       |
> | Ours(Virtual)  | 89.47   | 67.13      | +0.0025 G       |
>
> Table Note: Performance on Waterbirds on ViT-B/16 backbones. Efficiency are reported as FLOPs *increments* relative to CLIP baseline. Avg. and Worst. indicates average accuracy and the worst group accuracy.
>
> We acknowledge that Appendix E.1 did not clearly reference and we will update the draft to clarify table 13-14 indices and improve formatting for better readability.
>
> $ $
>
> **Q5: Justification for COCO-GB performance**
>
> **A5:** Thank you for raising this point. On COCO-GB v1 we report two key metrics: **average accuracy** which measures overall recognition performance and utility; **gender gap**, which indicates fairness. In Tab.8, our virtual variant achieves **the lowest gap of 0.8%** (compared to CLIP’s 6.2%) while still reaching 91.6 % average accuracy, **outperforming all other methods on both fairness and utility simultaneously.**
>
> A further advantage of our framework is its **tunable suppression factor $\hat{\lambda}$** . Users who prefer higher accuracy over strict parity (or vice-versa) can adjust $\hat{\lambda}$ to smoothly trade off between average accuracy and gap, making the method adaptable to different deployment requirements. We will clarify these points in the revised draft.
>
> $ $
>
> **Q6: Discussion on multiple models approach**
>
> **A6:** Thank you for thoughtful suggestion. Ensemble methods that aggregate predictions from multiple models or strategies are indeed effective in bias mitigation. For instance, Li .et al [5] combines classifiers trained on various augmentation models (e.g., watermark, texture, background) and aggregates their outputs to improve robustness. However, this approach requires retraining multiple specialized models, making it unsuitable for zero-shot settings. To provide a fairer comparison, we included the counterfactual variant of AugMix, which uses various image transformation strategies similar to [5].
>
>  Additionally, we implemented the **zero-shot ensemble strategy** proposed in [6]. This approach aggregates multiple pre-trained VLMs (e.g., CLIP variants) without fine-tuning, improving over any single model by leveraging architectural diversity. We evaluate these on benchmarks and present results below.
>
> | Method         | Waterbirds  |            | COCO-GB     |                  |      |
> | -------------- | ----------- | ---------- | ----------- | ---------------- | ---- |
> |                | **Avg. ↑** | **Worst. ↑** | **Avg. ↑** | **Gender Gap ↓** |      |
> | AugMix         | 72.54       | 52.28      | 90.00       | 1.60             |      |
> | $ZN_{E_n}$ [6]  | 77.32       | 54.86      | 90.80       | 2.80             |      |
> | Ours(External) | 86.40       | 70.87      | 91.10       | 2.60             |      |
> | Ours(Internal) | 85.43       | 82.71      | 91.25       | 1.50             |      |
> | Ours(Virtual)  | 89.47       | 67.13      | 91.40       | 0.80             |      |
>
> Table Note: Performance on ViT-B/16 backbones.
>
> **Reference:**
>
> [1] Sauer A, Geiger A. "Counterfactual generative networks". arXiv 2021.
>
> [2] Nam J, Cha H, Ahn S. "Learning from failure: De-biasing classifier from biased classifier". NIPS 2020.
>
> [3] Zhang, Marvin, Sergey Levine, and Chelsea Finn. "Memo: Test time robustness via adaptation and augmentation." NeurIPS 2022.
>
> [4] Dunlap et al., “Diversify your vision datasets with automatic diffusion-based augmentation,” NeurIPS 2023.
>
> [5] Li, Zhiheng, et al. "A whac-a-mole dilemma: Shortcuts come in multiples where mitigating one amplifies others." CVPR2023.
>
> [6] Lu, Zhihe, et al. "Beyond sole strength: Customized ensembles for generalized vision-language models." ICML2024.

---

> > ### Comment · Reviewer_XSqm · 2025-08-05
> > **Thanks for the rebuttal, few follow up questions**
> >
> > Overall, I find the rebuttal answers most of my questions. I have a few questions:
> >
> > For the question about contextual bias, the authors state that Empirical evidence exists that context drives most failures. Could the authors provide the exact references to the empirical findings?
> >
> > Additionally, for the claim about 'Better zero-shot generalization', are there any empirical evidences for this? While I agree that the texture and shape models may introduce target-specific information, and limit novel categories. I am interested to see how context is useful in such scenarios. Would the authors point to the specific experiment from their paper that highlights this aspect?
> >
> > For the experimental details on sample size, I recommend including this in the appendix, not just the codebase.
> >
> > Thanks for including ALIA [4], which is stronger than Memo [1], due to the usage of diffusion models for more domain diversity. The performance of ALIA [4] reported by the authors is significantly higher than the original work. Hence, I recommend including the detailed experimental details for this.

---

> > > ### Author Response · Authors · 2025-08-05
> > >
> > > Thank you very much for the thoughtful follow-up questions. We are glad to address each point in turn:
> > >
> > > **1. Regarding empirical evidence of contextual bias driving most model failures**
> > >
> > > Empirical studies have shown that VLMs like CLIP are vulnerable to harmful contextual biases embedded in their pretraining data. For instance, *Shtedritski et al.*[1] proved that simply adding a red circle around an object (i.e. a simple form of visual context manipulation) can misdirect CLIP’s attention and lead to incorrect predictions, even when target object remains visible. Their findings hold across both large-scale (LAION-2B) and smaller-scale (YFCC15M, where red circles are rare) pretraining dataset. Notably, ​**the severity of misclassification increases with model scale**​, aligning with observation by *Kaiming et al.*[2] that larger models are more prone to shortcut learning.
> > >
> > > In contrast, CLIP exhibits strong zero-shot performance [3] on texture-focused datasets (e.g., DTD2, FWD) and tasks involving animals' shape/texture cues, **outperforming texture-specific models, especially as model capacity increases**. This contrast highlights that for vision-language models, particularly those with large parameter counts, ​contextual bias remains a dominant factor contributing to prediction errors. We will clarify these evidence in related work section.
> > >
> > > **2. Regarding experimental evidence for "Better zero-shot generalization"**
> > >
> > > PerceptionCLIP [4] introduces an indirect experimental approach to quantitatively investigate how different cues affect generalization. By appending different types of suffixes to textual prompts, the study reveal how CLIP leverages these signals during inference. Building on this idea, we can further designs interventions that deliberately perturb suffix types (e.g., adding random , misleading or correct suffixes) and **compares the resulting posterior probabilities and worst-group accuracies under different prompting strategies**, such as those related to context or to texture/shape.
> > >
> > > In our paper, the theoretical analysis (Appendix B.5) and qualitative results (e.g., Fig. 8) further illustrate how misleading or inappropriate contextual information can inflate hallucination scores. This reinforces the importance of properly modeling contextual effects to ensure safer and more reliable generalization. We will clarify these in Appendix B.5.
> > >
> > > $ $
> > >
> > > **3. Clarification on experimental configuration**
> > >
> > > We agree with the reviewer’s suggestion to clarify implementation details. We will **add a summary table of key hyperparameters** in Appendix C.2, and ensure these are explicitly referenced in the main text.
> > >
> > > $ $
> > >
> > > **4. Differences compared to the original ALIA paper**
> > >
> > > To ensure fair comparison with other baselines in the table above, **we used ViT-B/16 pretrained on LAION-2B, rather than the ResNet-50 pretrained on ImageNet used in original ALIA paper**. This explains the higher accuracy reported in our implementation. Moreover, we **preserved all other configuration details of ALIA** (e.g., seven prompts corresponding to augmented sample size of M=7, the InstructPix2Pix editing method, and semantic prompt filtering) as described in their released codebase. We will clarify these settings in the revised paper and release our reproduction code.
> > >
> > > Thank you again for your insight comments. If you have any further suggestions, please let us know and your feedback are truly valuable to us.
> > >
> > > **Reference:**
> > >
> > > [1] Shtedritski, et al. "What does clip know about a red circle? visual prompt engineering for vlms." ICCV 2023.
> > >
> > > [2] Kaiming He, et al. "A decade’s battle on dataset bias: Are we there yet?" ICLR 2025.
> > >
> > > [3] Wu C, et al. "How well does CLIP understand texture?" ECCV 2022.
> > >
> > > [4] An, Bang, et al. "PerceptionCLIP: Visual classification by inferring and conditioning on contexts." ICLR 2024.

---

> > > > ### Comment · Reviewer_XSqm · 2025-08-05
> > > > **Post follow-up questions**
> > > >
> > > > Thank you for your response.
> > > > The referenced works suggest that contextual features may contribute to model failures, particularly as model capacity increases. Including the above discussion of PerceptionCLIP [15] in the appendix would be beneficial, as stated. I also appreciate the clarification regarding ALIA (Dunlap et al., 2023), which helped resolve my doubt about the significant increase in performance.

---

> > > > > ### Author Response · Authors · 2025-08-05
> > > > > **Thank you for the helpful feedback**
> > > > >
> > > > > Thank you for your valuable suggestions. We're glad to have addressed your concerns. We will carefully incorporate your recommendations and add the relevant clarifications in the revised version.

---

### Comment · Area_Chair_WRHo · 2025-08-05

Dear Reviewers,

Thanks for your efforts reviewing the papers. Since we have fewer than two days before the author-reviewer discussion period ends (August 6, 11:59pm AoE), could you please do the following ASAP: carefully read the other reviews and the authors' rebuttal, and reply to the authors regarding whether your questions or concerns have been addressed. If any issues remain unresolved, please specify them so that the authors have a fair opportunity to respond before the author-reviewer window closes.

If you have done so, please ignore this comment.

Best,
AC

---

### Decision · Program_Chairs · 2025-09-17

**Decision:**

Accept (poster)

**Comment:**

The final scores for this submission are 5/5/4/4. After the rebuttal, the reviewers consistently agreed that the authors had addressed their concerns and gave a positive attitude toward the work, highlighting its satisfactory novelty, moderate resource requirements, and solid theoretical explanations. After carefully reviewing both the paper and the rebuttal, I recommend accepting this submission.